# COSMOS: A Hybrid Adaptive Optimizer for Efficient Training of Large Language Models

**Liming Liu**
Georgia Tech

**Zhenghao Xu**
Georgia Tech

**Zixuan Zhang**
Georgia Tech

**Hao Kang**
Georgia Tech

**Zichong Li**
Georgia Tech

**Chen Liang**
Microsoft

**Weizhu Chen**
Microsoft

**Tuo Zhao**
Georgia Tech

## Abstract

Large Language Models (LLMs) have demonstrated remarkable success across various domains, yet their optimization remains a significant challenge due to the complex and high-dimensional loss landscapes they inhabit. While adaptive optimizers such as AdamW are widely used, they suffer from critical limitations, including an inability to capture interdependencies between coordinates and high memory consumption. Subsequent research, exemplified by SOAP, attempts to better capture coordinate interdependence but incurs greater memory overhead, limiting scalability for massive LLMs. An alternative approach aims to reduce memory consumption through low-dimensional projection, but these methods lose the gradient information in the residual space, resulting in less effective optimization. In this paper, we propose COSMOS, a novel hybrid optimizer that leverages the varying importance of eigensubspaces in the gradient matrix to achieve memory efficiency without compromising optimization performance. The design of COSMOS is motivated by our empirical insights and practical considerations. Specifically, COSMOS applies SOAP to the leading eigensubspace, which captures the primary optimization dynamics, and MUON to the remaining eigensubspace, which is less critical but computationally expensive to handle with SOAP. This hybrid strategy significantly reduces memory consumption while maintaining robust optimization performance, making it particularly suitable for massive LLMs. Numerical experiments on various datasets and transformer architectures are provided to demonstrate the effectiveness of COSMOS.

## 1 Introduction

The optimization of Large Language Models (LLMs) is fundamental to their success, enabling these models to achieve state-of-the-art performance across diverse tasks. However, the non-convex loss landscapes inherent to LLMs, which can contain hundreds of billions or even trillions of parameters (Achiam et al., 2023), present significant optimization challenges. Adaptive optimizers, such as Adam (Kingma, 2014) and its variants AdamW (Loshchilov, 2017), have emerged as de facto standards due to their ability to dynamically adjust learning rates based on the second moment of the gradient. Despite their widespread adoption, these methods suffer from two critical limitations that impede their effectiveness and scalability in the context of increasingly large and complex LLMs:

(I) Adam and its variants have limitations in adaptive learning rates. By adjusting learning rates independently for each parameter, the method reduces computational complexity but may not fully capture parameter interdependencies. In complex architectures of LLMs, this independent approach can lead to suboptimal parameter updates (Zhang et al., 2024a).

(II) Another limitation of Adam and its variants lies in the substantial memory requirement for storing per-parameter adaptive learning rates and gradient statistics. As LLM sizes increase, memory consumption becomes prohibitively large, impeding scalability.

Correspondence to: `lliu606,zhenghaoxu,zzhang3105,tourzhao@gatech.edu`

To address the limitations of Adam and its variants, researchers have pursued two main approaches. The first approach, exemplified by algorithms such as Shampoo (Gupta et al., 2018) and the more recent SOAP (Vyas et al., 2024), employs sophisticated techniques to capture curvature information and parameter interdependencies. These methods utilize rotational matrices, derived through (approximate) singular value decomposition (SVD) of the gradient matrix, to provide a more comprehensive representation of the loss landscape's geometry. This approach allows for a better approximation of the full preconditioning matrix, enabling the capture of inter-coordinate dependencies. However, the improved capability of representing parameter interactions comes at the cost of substantial computational and memory overhead, rendering these algorithms challenging to implement for large-scale LLMs, where memory efficiency is crucial.

The second approach focuses on reducing memory consumption through various approximation techniques. Algorithms such as AdaFactor (Shazeer and Stern, 2018) and Adam-mini (Zhang et al., 2024b) aim to decrease memory usage by approximating the second moment of the gradient matrix. Adam-mini employs a component-specific approach, averaging second moments neuron-wise for certain layers. Meanwhile, AdaFactor utilizes a rank-1 approximation of the second moments. While these methods reduce memory cost, their approximations oversimplify the structure of the gradient matrix's second order moments, compromising optimization performance. The trade-off between memory efficiency and the preservation of gradient statistics remains a crucial challenge.

More recent approaches, such as GaLore (Zhao et al., 2024a) and MUON (Jordan et al., 2024), have attempted to strike a balance between computational complexity, memory consumption, and optimization performance in LLM training. GaLore, which can be viewed as a memory-efficient variant of SOAP, approximates the first and second moments of the gradient matrix in the leading eigensubspace. While it effectively reduces memory consumption, Liang et al. (2024) find that its effectiveness diminishes for sequence lengths exceeding 256. MUON, essentially an approximation of Shampoo based on Newton-Schulz transformation proposed in Bernstein and Newhouse (2024), aims to decrease computational complexity. However, MUON only estimates the eigensubspaces based on the gradient on one batch, rather than capturing the comprehensive distribution of gradients across the entire optimization process.

In this paper, we propose COSMOS, a novel hybrid optimizer that addresses the limitations of existing methods by exploiting the varying importance of eigensubspaces in the gradient matrix. Our approach decomposes the gradient into two parts: a projection onto the leading eigensubspace and a projection onto the remaining eigensubspace. The leading eigensubspace captures the most significant directions of change in the gradient, typically corresponding to the most important optimization dynamics. For this part, we apply a SOAP-like optimization strategy. However, by crucially restricting SOAP to the leading eigensubspace, COSMOS only needs to maintain the projection matrix and the second-order moment within this small subspace, thereby retaining SOAP's ability to capture parameter interdependencies while substantially lowering its memory cost. The remaining eigensubspace, while less critical, still significantly influences optimization performance. To address this, we employ MUON as a more efficient alternative to SOAP for this high-dimensional space. Such a hybrid approach allows COSMOS to maintain optimization effectiveness while significantly reducing memory requirements compared to SOAP, potentially enabling the training of larger LLMs or the use of increased batch sizes.

We highlight the key contributions of this paper as follows: **(1)** We propose a novel hybrid optimization strategy. This leads us to develop the COSMOS algorithm, which synergizes the strengths of SOAP and MUON by decomposing the gradient matrix into eigensubspaces of varying importance. **(2)** COSMOS achieves significant memory consumption reduction compared to the SOAP algorithm, while achieving equally or better optimization performance.

## 2  RELATED WORK

The optimization of LLMs has seen significant advancements in recent years, with various approaches aimed at improving efficiency and performance. This section discusses key related works in adaptive optimization, memory-efficient techniques, and specialized algorithms for LLMs.

**Coordinate-wise adaptive optimizers:** Adam (Kingma, 2014) and AdamW (Loshchilov, 2017) have become standards in deep learning optimization due to their ability to dynamically adjust

learning rates based on the first and second moments of the gradients. However, these methods treat parameters independently, failing to capture interdependencies between coordinates. This limitation can lead to suboptimal updates, especially in the complex architectures of LLMs. Other adaptive optimizers such as Lion (Chen et al., 2023), Sophia (Liu et al., 2023), and Adafactor (Shazeer and Stern, 2018; Zhai et al., 2022) have shown comparable performance to AdamW in LLM pretraining but have not significantly surpassed it, suggesting the need for non-diagonal preconditioners.

**Second-Order Optimizers:** Researchers have explored second-order optimization techniques for training large models. These methods can be broadly categorized into Hessian-free approaches and Hessian estimation methods. Hessian-free methods, such as those proposed by Martens (2010) and Martens and Grosse (2015), optimize without explicitly computing the Hessian matrix. On the other hand, Hessian estimation methods maintain an efficient approximation of the Hessian for neural networks. Notable examples include KFAC (Martens and Grosse, 2015), Shampoo (Gupta et al., 2018) and SOAP (Vyas et al., 2024).

⋄ *Shampoo and Its Variants:* Shampoo (Gupta et al., 2018), another second-order optimization algorithm, is motivated by the online learning algorithm Adagrad (Duchi et al., 2011). Shampoo also employs a layer-wise Kronecker-factored preconditioner. A recent distributed implementation of Shampoo (Shi et al., 2023) won an optimization efficiency benchmark (Dahl et al., 2023), highlighting the practical utility of second-order methods in deep learning. Other works (Anil et al., 2020; Peirson et al., 2022; Lin et al., 2024; Wang et al., 2024; Zhao et al., 2024b) have proposed various strategies to improve Shampoo's scalability.

⋄ *SOAP:* SOAP algorithm (Vyas et al., 2024) establishes a formal connection between Shampoo and Adafactor. SOAP is equivalent to running Adafactor in the eigenbasis of Shampoo's preconditioner, leading to a simpler and computationally efficient algorithm. By continually updating the running average of the second moment in the current (slowly changing) coordinate basis, SOAP mitigates the performance degradation associated with less frequent eigendecomposition computations. SOAP has shown significant improvements over AdamW in per-token efficiency.

**Memory-efficient optimizers:** As LLM sizes increase, memory efficiency becomes crucial. Several approaches have been proposed to reduce the memory footprint of optimizers:

⋄ *Adafactor and Adam-mini:* Shazeer and Stern (2018) use a low-rank approximation of the second moments to reduce memory consumption. It has been widely used in LLMs due to memory efficiency. Zhang et al. (2024b) achieve comparable performance than AdamW with a 50% smaller memory footprint. It reduces memory by carefully partitioning parameters into blocks and assigning a single learning rate to each block based on the Hessian structure of neural networks.

⋄ *GaLore:* Zhao et al. (2024a) reduce Adam's memory footprint by maintaining momentum in a low-rank subspace derived from the singular value decomposition (SVD) of the gradients. However, its effectiveness diminishes for sequence lengths exceeding 256, as shown in Liang et al. (2024).

⋄ *MUON:* The MUON optimizer (Jordan et al., 2024) can be viewed as an efficient approximation of Shampoo. It employs a Newton-Schulz transformation to approximately implement the Kronecker-factored preconditioner. While computationally more complex than Adam, MUON only adds minor overhead to the overall training time due to efficient parallelization of matrix operations.

These advancements highlight the efforts to improve the training efficiency and performance of LLMs. However, each approach comes with its own trade-offs in terms of computational complexity, memory requirements, and performance. Our work builds upon these insights to develop a hybrid approach that aims to balance these factors effectively, combining the strengths of different methods to achieve both memory efficiency and robust optimization performance for massive LLMs.

## 3 COSMOS: A HYBRID ADAPTIVE OPTIMIZER

We present a novel hybrid optimizer – COSMOS in Algorithm 1, which can achieve memory efficiency without compromising performance for training LLMs. Without loss of generality, we use $m$ and $n$ to denote the numbers of rows and columns in a $m$ by $n$ matrix, and we assume $m > n$. For simplicity, we use the following notations:

- Matrix Sign Operator: Given a matrix $X \in \mathbb{R}^{m \times n}$ and its reduced-SVD $X = UDV^\top$, where $D \in \mathbb{R}^{n \times n}$ is a diagonal matrix containing all singular values of $X$, and $U \in \mathbb{R}^{m \times n}$ and $V \in \mathbb{R}^{n \times n}$ are left and right singular vector matrices, respectively. We define

$$\mathtt{MatSgn}(X) = UV^\top.$$

- Newton Schulz (NS) transformation: Given a matrix $X_0 \in \mathbb{R}^{m \times n}$, where $\|X_0\|_\mathrm{F} \leq 1$, we define

$$\mathtt{NS5}(X_0) = X_5.$$

where $X_5$ is obtained by $X_{k+1} = aX_k + bX_k X_k^\top X_k + cX_k X_k^\top X_k X_k^\top X_k$ for $k = 0, 1, ..., 4$ with $a = 3.4445$, $b = -4.7750$ and $c = 2.0315$. Bernstein and Newhouse (2024) first mentioned this transformation to approximate the matrix sign operator without specifying the coefficient. Jordan et al. (2024) later used an ad-hoc gradient based approach to find the set of coefficients here.

- Normalization operator: $\mathtt{NORM}(X) = \sqrt{n}X/\|X\|_\mathrm{F}$, where $\|\cdot\|_\mathrm{F}$ denotes the Frobenius norm. The normalization operator is used to normalize the output of the NS transformation.

- Gram–Schmidt procedure: $\mathtt{QR}(X)$.

---

**Algorithm 1** COSMOS for an $m \times n$ layer $W$. Per layer, we maintain four matrices: $U \in \mathbb{R}^{n \times r}, S \in \mathbb{R}^{r \times r}, V \in \mathbb{R}^{m \times r}$ and $M \in \mathbb{R}^{m \times n}$.

**input** Learning rate $\eta$, combination weight $\gamma$, projection rank $r \ll n$, momentum parameters $(\beta_1, \beta_2)$, perturbation parameter $\epsilon$. For simplicity, we omit the initialization.

1: **for** $t = 0, ...$ **do**
2:     Sample batch $\mathcal{M}_t$
3:     $G_t \leftarrow \nabla_W \phi_{\mathcal{M}_t}(W_t)$
4:     $M_t \leftarrow \beta_1 M_{t-1} + (1 - \beta_1)G_t$
5:     $U_t \leftarrow \mathtt{QR}(\beta_2 U_{t-1}S_{t-1} + (1 - \beta_2)G_t^\top G_t U_{t-1})$
6:     $S_t \leftarrow U_t^\top(\beta_2 U_{t-1}S_{t-1}U_{t-1}^\top + (1 - \beta_2)G_t^\top G_t)U_t$
7:     $V_t \leftarrow \beta_2 V_{t-1} + (1 - \beta_2)(G_t U_t) \odot (G_t U_t)$
8:     $A_t = \left( \dfrac{M_t U_t/(1 - \beta_1^t)}{\sqrt{(V_t + \epsilon)/(1 - \beta_2^t)}} \right) U_t^\top$
9:     $B_t \leftarrow \mathtt{NORM}\left( \mathtt{NS5}\left( \dfrac{M_t - M_t U_t U_t^\top}{\|M_t - M_t U_t U_t^\top\|_\mathrm{F}} \right) \right)$
10:     $\tilde{G}_t \leftarrow A_t + \gamma \cdot B_t \cdot \sqrt{m}$
11:     $W_{t+1} \leftarrow W_t - \eta \cdot \mathtt{NORM}(\tilde{G}_t) \cdot \sqrt{m}$
12: **end for**

---

**Design principle** The design of COSMOS is guided by a simple principle: instead of maintaining SOAP's full second-moment matrix—which is memory-prohibitive—we track its dominant eigenspace and operate in a low-dimensional subspace.

In the SOAP algorithm, the exponential moving average (EMA) of the second moment is

$$H_t = \beta_2 H_{t-1} + (1 - \beta_2)G_t^\top G_t, \tag{1}$$

where $G_t$ is the stochastic gradient. Because $H_t \in \mathbb{R}^{n \times n}$ is dense, storing it is infeasible for large $n$. COSMOS avoids this by maintaining (i) an orthonormal basis $U_t \in \mathbb{R}^{n \times r}$ for the leading eigenspace of $H_t$ and (ii) a projected second-moment matrix $S_t \in \mathbb{R}^{r \times r}$ with

$$S_t \approx U_t^\top H_t U_t.$$

Assume at step $t - 1$ that $U_{t-1}$ spans the dominant eigenspace and $S_{t-1} \approx U_{t-1}^\top H_{t-1} U_{t-1}$. Approximating $H_{t-1}$ by its rank-$r$ surrogate, $U_{t-1}S_{t-1}U_{t-1}^\top$, and substituting into Equation (1) yields

$$\tilde{H}_t = \beta_2 U_{t-1}S_{t-1}U_{t-1}^\top + (1 - \beta_2)G_t^\top G_t.$$

We then update the basis via a one-step power iteration:

$$U_t = \mathtt{QR}(\tilde{H}_t U_{t-1}),$$

and refresh the projected second moment by

$$S_t = U_t^\top \tilde{H}_t U_t.$$

These two steps track the dominant eigenspace and its curvature information with $O(nr)$ memory.

Given $U_t$, Line 7 of Algorithm 1 maintains the EMA of the projected gradients $V_t \in \mathbb{R}^{m \times r}$, and Line 8 performs a SOAP-like adaptive update within the subspace spanned by $U_t$, producing $A_t$ after projecting back to the full parameter space. This is the SOAP component of COSMOS.

To complement the low-rank update, COSMOS applies a MUON-inspired preconditioner on the orthogonal complement of $U_t$. Writing the orthogonal projector as $P_t^\perp = I - U_t U_t^\top$, Line 9 forms

$$B_t = \text{NORM}\left( \text{NS5}\left( \frac{M_t P_t^\perp}{\|M_t P_t^\perp\|_\text{F}} \right) \right) = \text{NORM}\left( \text{NS5}\left( \frac{M_t - M_t U_t U_t^\top}{\|M_t - M_t U_t U_t^\top\|_\text{F}} \right) \right), \tag{2}$$

where NS5 applies directly to the residual momentum; no additional matrices are stored.

Finally, Lines 10–11 combine the two components:

$$\tilde{G}_t = A_t + \gamma B_t \sqrt{m}, \qquad W_{t+1} = W_t - \eta \, \text{NORM}(\tilde{G}_t) \sqrt{m}.$$

The normalization ensures the update has Frobenius norm $\Theta(\sqrt{mn})$, matching MUON's scaling. In sum, COSMOS adaptively preconditions the leading eigenspace as in SOAP while using MUON on the residual, achieving robust optimization with substantially reduced memory.

---

**Algorithm 2** (One-side) SOAP

**input** Learning rate $\eta$, momentum parameters $(\beta_1, \beta_2)$, perturbation parameter $\epsilon$.
1: **for** $t = 0, \dots$ **do**
2:     Sample batch $\mathcal{M}_t$
3:     $G_t \leftarrow \nabla_W \phi_{\mathcal{M}_t}(W_t)$
4:     $M_t \leftarrow \beta_1 M_{t-1} + (1 - \beta_1) G_t$
5:     $L_t \leftarrow \beta_2 G_t^\top G_t U_{t-1} + (1 - \beta_2) G_t^\top G_t U_{t-1}$
6:     $U_t \leftarrow \text{QR}(L_t U_{t-1})$
7:     $G_t' \leftarrow G_t U_t$
8:     $V_t \leftarrow \beta_2 V_{t-1} + (1 - \beta_2)(G_t' \odot G_t')$
9:     $A_t = \left( \dfrac{M_t U_t / (1 - \beta_1^t)}{\sqrt{(V_t + \epsilon)/(1 - \beta_2^t)}} \right) U_t^\top$
10:    $W_{t+1} \leftarrow W_t - \eta A_t$
11: **end for**

**Algorithm 3** MUON

**input** Learning rate $\eta$, momentum parameters $\mu$.
1: **for** $t = 0, \dots$ **do**
2:     Sample batch $\mathcal{M}_t$
3:     $G_t \leftarrow \nabla_W \phi_{\mathcal{M}_t}(W_t)$
4:     $M_t \leftarrow \mu M_{t-1} + G_t$
5:     $N_t \leftarrow \mu M_t + G_t$
6:     $B_t \leftarrow \text{NS5}(N_t / \|N_t\|_\text{F})$
7:     $W_{t+1} \leftarrow W_t - \eta B_t \cdot \sqrt{m}$
8: **end for**

---

**Remark 1** *As can be seen, COSMOS only needs to maintain four matrices in the memory: $M_t \in \mathbb{R}^{m \times n}$, $U_t \in \mathbb{R}^{n \times r}$, $S_t \in \mathbb{R}^{r \times r}$ and $V_t \in \mathbb{R}^{m \times r}$. In sharp contrast, even one-sided SOAP (2) needs to maintain $M_t \in \mathbb{R}^{m \times n}$, $L_t \in \mathbb{R}^{n \times n}$, $U_t \in \mathbb{R}^{n \times n}$ and $V_t \in \mathbb{R}^{m \times n}$. The resulting memory overhead is which is significantly larger than that of COSMOS.*

**Remark 2** *Recall that the the computation complexity of the QR decomposition on a matrix of the shape $n \times r$ is $O(nr^2)$ when $rn$, so the low rank QR decomposition of $\beta_2 U_{t-1} S_{t-1} + (1 - \beta_2) G_t^\top G_t U_{t-1}$ in COSMOS is actually very quick since $r \ll n$ (and much quicker than that in SOAP, which is $O(n^3)$). Therefore, unlike SOAP which needs to consider the preconditioning frequency for performing QR decomposition, we can carry out QR decomposition at every step with virtually no overhead. In addition, PyTorch provides an efficient implementation of QR method, which is also used by SOAP. In Table 6, we provide the comparison of wall-clock time per iteration to show that compared to MUON, COSMOS only incurs a very slight increase in wall-clock time.*

## 3.1 MEMORY USAGE COMPARISON

For comparison, we list the memory usage of the optimization states in Adam, Adam-mini, SOAP, MUON and COSMOS for training transformer models in Table 1. For simplicity, we assume that the attention weight matrices $W_Q, W_K, W_V, W_O \in \mathbb{R}^{d \times d}$ and the MLP weight matrices $W_1 \in \mathbb{R}^{d \times 4d}$

Table 1: Memory usage of the optimization states in different algorithms for training transformers.

| Adam | Adam-mini | SOAP | MUON | COSMOS |
|------|-----------|------|------|--------|
| $24d^2$ | $12d^2$ | $66d^2$ | $12d^2$ | $13d^2$ |

and $W_2 \in \mathbb{R}^{4d \times d}$. Note that in practical LLMs, the dimensionalities of $W_1$ and $W_2$ might slightly vary. Moreover, we assume that the rank of the projection is $r = 0.05d$ for COSMOS.

We remark that Table 1 only compares the optimization states. In practice, however, besides the optimization states, the overall memory usage also includes the model weights and intermediate variables used in the forward and backward passes as well as additional memory overhead of I/O and computation. Therefore, we present a more detailed and practical memory profiling for training LLaMA-1B model in Section 4.

## 4 EXPERIMENTS

We evaluate the performance of COSMOS on pre-training various sizes of LLMs, in comparison with baseline algorithms including Adam (Kingma, 2014), Adam-mini (Zhang et al., 2024b), Ga-Lore (Zhao et al., 2024a), SOAP (Vyas et al., 2024) and MUON (Jordan et al., 2024). Note that for SOAP, MUON and COSMOS, the embedding and output weights are trained by Adam.

**Models and datasets.** We train LLaMA-type models (Touvron et al., 2023) on the C4 dataset (Raffel et al., 2020), which is a colossal, cleaned version of Common Crawl's web crawl corpus for pre-taining. We conduct comprehensive experiments and ablation studies on 130M models and demonstrate the token efficiency of COSMOS. We then scale up to 350M and 1B models to showcase the memory efficiency and small computational overhead of COSMOS. Due to limited computational resources, experiments on these larger models are less comprehensive, while still capable of illustrating the efficacy of our method. We train for one epoch on a portion of the C4 dataset, ranging from 5B to 26B tokens, and scaling with the model size according to the scaling law (Kaplan et al., 2020). We set the maximum sequence length as 1024 by default.

Besides LLaMA models and C4 dataset, we also conducted experiments with modded-NanoGPT (Jordan et al., 2024) on FineWeb (Penedo et al., 2024) and GPT-2 (Radford et al., 2019) on WikiText-103 (Merity et al., 2016) to evaluate the effectiveness of COSMOS across different settings.

### 4.1 COMPARISON ON LLAMA-130M

For LLaMA-130M, we train on a 5B subset of the C4 dataset. We compare COSMOS's validation loss with that of Adam, Adam-mini, GaLore, MUON, and SOAP.

**Hyperparameters.** For each method, we tune the corresponding learning rate to obtain optimal performance. We select the rank $r = 64$ for COSMOS and $r = 256$ for GaLore. To avoid multiple hyperparameter tuning, we set the discount factor in COSMOS as $\gamma = \eta/\eta_0$, where $\eta_0$ is the learning rate of Adam for training the embedding and output weights in the implementation of COSMOS. Other hyperparameter choices follow Zhao et al. (2024a) and are provided in detail in Section A.1.1.

**Main results.** We plot the validation loss curves in Figure 1. As illustrated, COSMOS consistently outperforms MUON with better stability and is comparable to SOAP. This showcases that our hybrid approach leveraging the leading eigensubspace captures the most important information for efficient update, allowing COSMOS to achieve similar per-token efficiency as SOAP. In addition, all three methods outperform the vanilla Adam and are much better than Adam-mini and GaLore, validating that inter-coordinate dependence is crucial for efficient optimization. We report the final validation perplexity in Table 2.

**Ablation on learning rates.** We experiment with different learning rates while keeping the rank $r = 64$ and discount factor $\gamma = \eta/\eta_0$ for COSMOS. As shown in the Table 3, COSMOS is not very sensitive to the learning rate, and it achieves the best performance at 5e-4. As a comparison, MUON is more sensitive to the learning rate, and it underperforms COSMOS across all learning rates.

**Ablation on rank and discount factor.** We also experiment with different ranks $r$ and discount factors $\gamma$ while keeping the learning rate as 5e-4 for COSMOS, and the results are summarized in Table 5. As illustrated, COSMOS is not very sensitive to $r$ and $\gamma$, and the best discount factor is

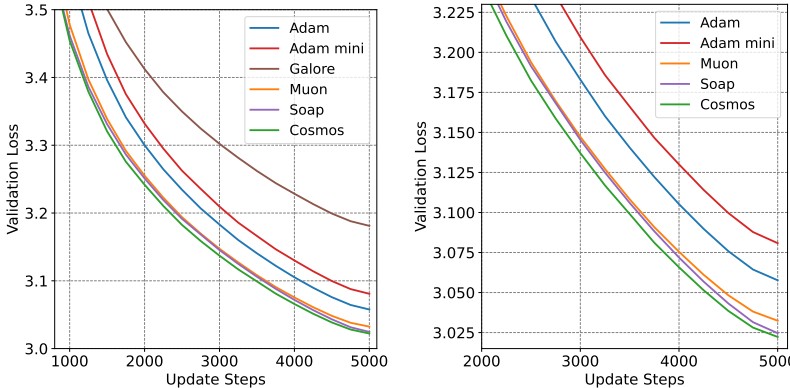

Figure 1: Performance on LLaMA-130M. COSMOS consistently outperforms baseline methods. In the right plot, we hide GaLore to better compare the performance of COSMOS with SOAP and MUON, as the curves are close in the left plot.

around $0.25$ to $0.5$ across different ranks. In practice, our choice $\gamma = \eta/\eta_0$ falls in this range, so it serves as a valid heuristic that prevents extra tuning of $\gamma$. We also conduct experiments with $\gamma = 0.25$ fixed while varying the learning rate. The results show that, when $\gamma = 0.25$, COSMOS consistently outperforms Muon across different learning rates, as shown in Table 4. We provide details in Section A. Moreover, we observe that as rank increases, COSMOS performs slightly worse and is more sensitive to the choice of $\gamma$. One possible explanation is that when the rank is large, the top-$r$ eigenvalues of $M_t$ contain some smaller values that are close to the remaining eigenvalues. For these eigenvalues, the one-step power iteration (Line 5 in Algorithm 1) cannot accurately approximate their corresponding eigensubspaces, leading to larger approximation errors and worse performance.

Table 2: Validation perplexity after training on C4 dataset. We train for 5000 steps on 130M and 350M models and 13000 steps on 1B model. COSMOS achieves the best validation perplexity.

| Size(Tokens) | 130M(5B) | 350M(10B) | 1B(26B) |
|---|---|---|---|
| Adam | 21.28 | 17.28 | 12.97 |
| Adam-mini | 21.78 | 18.03 | - |
| GaLore | 24.07 | 19.03 | - |
| SOAP | 20.59 | 16.32 | - |
| MUON | 20.69 | 16.49 | 12.57 |
| COSMOS | **20.54** | **16.21** | **12.46** |

Table 3: Validation perplexity under different learning rates. We set $\gamma = \eta/\eta_0$ and $r = 64$ for COSMOS. Our method outperforms MUON across all learning rates.

| lr | 2e-4 | 5e-4 | 1e-3 | 2e-3 |
|---|---|---|---|---|
| MUON | 21.72 | 20.75 | 20.69 | 26.74 |
| COSMOS | **21.17** | **20.54** | **20.62** | **21.00** |

Table 4: Validation perplexity under different learning rates. We set $\gamma = 0.25$ and $r = 64$ for COSMOS. Our method outperforms MUON across all learning rates.

| lr | 2e-4 | 5e-4 | 1e-3 | 2e-3 |
|---|---|---|---|---|
| MUON | 21.72 | 20.75 | 20.69 | 26.74 |
| COSMOS | **21.09** | **20.54** | **20.57** | **20.87** |

**Effect of normalization.** COSMOS applies a normalization step (Line 9 in Algorithm 1) after the NS transformation compared to MUON (Algorithm 3). Empirically, we find that COSMOS also outperforms the normalized version of MUON (see Figure 7 in Section D.1). This implies that normalization is not the only driving force behind COSMOS's efficiency.

**GaLore degradation on long sequences.** In our experiment, we observe that GaLore performs much worse than COSMOS and other baselines, including Adam. Such a degradation is less sig-

Table 5: Valid perplexity of COSMOS under different $r$ and $\gamma$. COSMOS is not very sensitive to $r$ and $\gamma$, and consistently outperforms MUON (20.69) except for only one config ($r = 128$, $\gamma = 1$).

| $r\backslash\gamma$ | 0.1 | 0.25 | 0.5 | 1 |
|---|---|---|---|---|
| 32 | 20.58 | 20.55 | **20.54** | 20.54 |
| 64 | 20.62 | **20.54** | 20.57 | 20.61 |
| 128 | 20.65 | 20.58 | 20.63 | 20.72 |

nificant on the shorter sequences with length 256 (see Figure 10 in Section D.5, the setting adopted in the original GaLore paper (Zhao et al., 2024a). This observation of GaLore degradation on long sequences aligns with Liang et al. (2024), while a similar phenomenon appearing in fine-tuning is reported by Pan et al. (2024). In contrast, COSMOS consistently outperforms Adam from short to long sequences without suffering from degradation.

## 4.2 Scaling Up to LLaMA-350M and LLaMA-1B

To further illustrate the effiency of COSMOS, we scale up to larger models and more tokens. For LLaMA-350M, we train on a 10B subset of the C4 dataset for 5000 steps. We compare COSMOS with Adam, Adam-mini, GaLore, MUON, and SOAP. For LLaMA-1B, we train on a 26B subset for 13000 steps. Given limited GPU resources, we compare COSMOS with Adam and MUON. Adam serves as the standard baseline, while MUON achieves better performance than Adam while requiring less memory and little computational overhead. Although SOAP show superior performance among baselines, we exclude it from our comparison as its complete training for the 1B model exceeds our available resources. More experiment details are provided in Sections A.1.2 and A.1.3.

**Main results of token efficiency.** Figures 2a and 2b display the validation loss curves for the 350M and 1B models, and Table 2 presents their final validation perplexities. COSMOS demonstrates superior performance compared to all baselines across both model sizes, matching the results on the 130M model and showcasing consistent token efficiency across different model sizes.

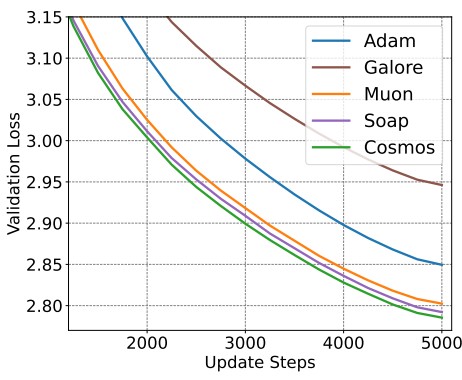
(a) LLaMA-350M trained on C4 dataset.

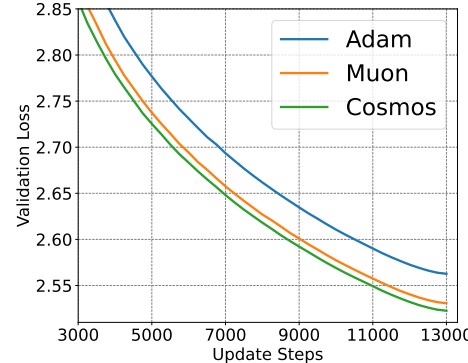
(b) LLaMA-1B trained on C4 dataset.

Figure 2: Comparison of performance on LLaMA-350M and LLaMA-1B trained on the C4 dataset.

**Memory and computation time profiling.** To illustrate the memory efficiency and small computation overhead of COSMOS, we conduct a profiling experiment on the 1B model. We fix the batch size as 10 and the gradient accumulation steps as 25 for all methods and record the maximum GPU memory usage and time spent during the entire forward-backward propagation and optimizer update process for one iteration. As shown in Table 6, COSMOS achieves much lower maximum GPU memory usage than Adam (**6.8%**) and SOAP (**19.4%**), with slightly more overhead compared to MUON. In terms of wall-clock time per iteration, COSMOS is comparable to MUON and is much better than SOAP. The fastest Adam method cannot achieve the same level of token efficiency as COSMOS. Therefore, COSMOS strikes a good balance between token and memory/computation overheads, achieving the best final perplexity at a much lower cost.

To better compare the methods in a practical setting, we evaluate the maximal batch size and throughput of COSMOS and baselines on a single NVIDIA A100 GPU with 80GB memory. The input sequence length is set to 1024. As shown in Table 7, COSMOS is **10.8%** faster than SOAP and comparable to MUON.

Table 6: GPU memory usage and wall-clock time per iteration on 1B model. We fix the batch size to be 10 for all methods. COSMOS has significantly less memory usage than SOAP and is comparable to memory-efficient methods like MUON, without introducing much computation overhead.

| Method | Memory | Wall-clock time |
|---|---|---|
| Adam | 62.75 G | **34.73** s |
| SOAP | 72.58 G | 39.51 s |
| MUON | **58.25** G | **35.56** s |
| COSMOS | **58.47** G | **35.75** s |

Table 7: System performance on single NVIDIA A100-80G GPU and corresponding throughput (number of samples processed per second on C4 dataset) of 1B model. Max batch size is defined as the maximum number of samples that fit within the GPU's memory capacity. Throughput is reported as the number of samples the GPU processes per second (samples/s).

| Method | Max batch size | Throughput(sample/s) |
|---|---|---|
| Adam | 13 | **7.24** |
| SOAP | 10 | 6.33 |
| MUON | **14** | **7.23** |
| COSMOS | **14** | **7.07** |

**Wall-Clock time plot for LLaMA-1B.** Based on the throughput we calculate in Table 7, we rescale the X-axis of Figure 2b to be wall-clock time and present the result in Figure 9 in Section D.4. Our results indicate that, in terms of actual training time, COSMOS still outperforms MUON and the Adam baseline, demonstrating COSMOS's potential for efficient pretraining.

## 4.3 ADDITIONAL EXPERIMENTS ON OTHER SETTINGS

Most current works on pretraining optimizers, due to limitations in resources and time, focus on a single model architecture and a single dataset — for example, GaLore (Zhao et al. (2024a),LLaMA on C4), SOAP (Vyas et al. (2024), OLMo (Groeneveld et al., 2024) on C4) and Muon (Jordan et al. (2024), Modded-NanoGPT on FineWeb). Our experiments on LLaMA with C4 are already aligned with these prior works, validating the performance of COSMOS. To demonstrate that COSMOS retains its advantages under other settings as well, however, we also conduct experiments in the following additional settings:

**LLaMA 130M on FineWeb:** To further assess the advantages of COSMOS over Muon across diverse settings, we conduct experiments with LLaMA under a substantially different configuration. Specifically, we train a 130M-parameter model on FineWeb-10B, corresponding to approximately 4× the data size recommended by Chinchilla scaling. We apply a weight decay of 0.1 to all optimizers. With the sequence length fixed at 1024, we increase the batch size to 4096. This setup features large batch size, large-scale data, and strong weight decay, making it closer to industrial-scale pretraining conditions. The results are shown in Figure 3.

**Modded-NanoGPT on FineWeb:** We also conduct experiments in Muon's original setting, namely Modified-NanoGPT on FineWeb. Since Muon has already performed extensive hyperparameter tuning in this setting, we do not tune Muon again but use the provided reproducible log. For COSMOS, we simply align the learning rate with that of Muon and follow other settings. We present the results in Figure 4. See Section B for an additional wall-clock time plot and the detailed configuration.

**GPT2 on Wikitext-103:** We also trained GPT2-small and GPT2-medium on WikiText-103 to compare COSMOS with Adam and Muon. In this setting, COSMOS still outperforms Muon and Adam. We present the result in Figure 5. See Section D.2 for the detailed configuration.

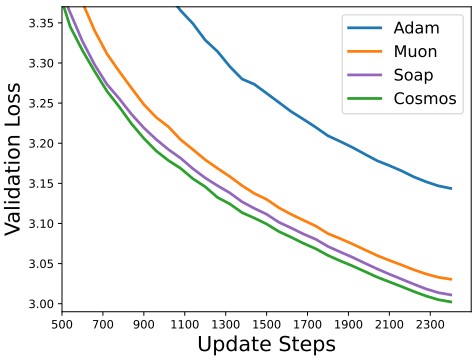

Figure 3: Comparison of optimization performance on 130M LLaMA trained on FineWeb dataset, with large batch size and data size. Weight decay is fixed at 0.1 for all optimizers.

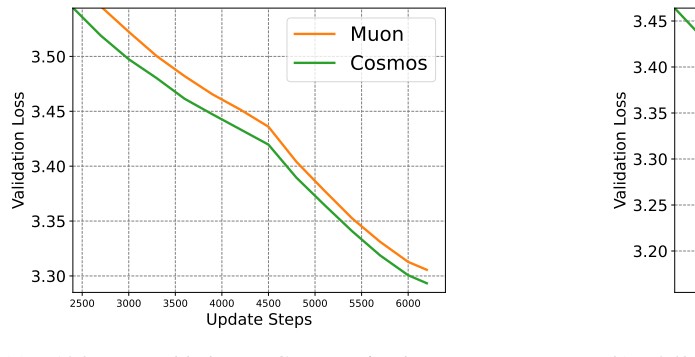

(a) 124M Modded-NanoGPT trained on FineWeb.

(b) 350M Modded-NanoGPT trained on FineWeb.

Figure 4: Comparison of optimization performance on 124M and 350M Modded-NanoGPT trained on the FineWeb dataset. COSMOS consistently outperforms MUON.

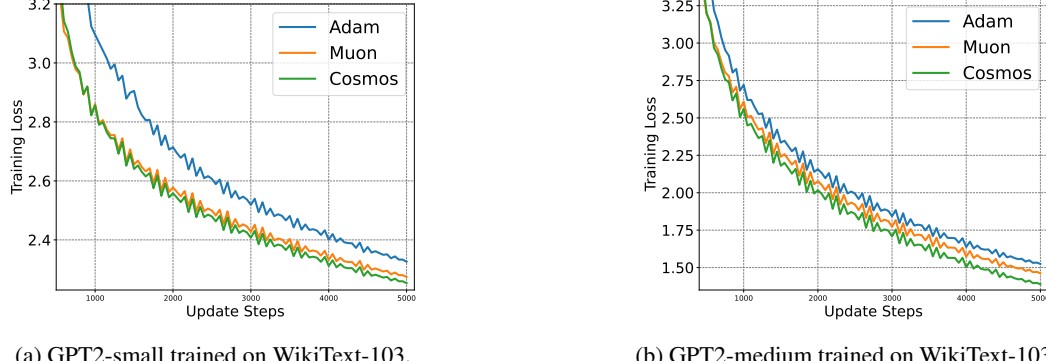

(a) GPT2-small trained on WikiText-103.

(b) GPT2-medium trained on WikiText-103.

Figure 5: Comparison of COSMOS, MUON and Adam on WikiText-103 using GPT2-small and GPT2-medium models. COSMOS consistently outperforms MUON and Adam.

## 5    CONCLUSION

We develop a hybrid adaptive optimizer, COSMOS, which leverages the varying importance of eigensubspaces in the gradient matrix to achieve token efficiency, memory efficiency, and high computation throughput simultaneously. By decomposing the gradient matrix into leading and remaining eigensubspaces and applying SOAP-like and MUON-like updates to them correspondingly, COSMOS uses significantly less memory than SOAP while achieving equal or better optimization performance. Comprehensive experiments show that COSMOS performs consistently well across different settings.

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

## A  EXPERIMENT DETAILS ON LLaMA MODELS

Many aspects of our setup such as models are the same as in Zhao et al. (2024a). We train language models on C4 tokenized with the T5 tokenizer (Raffel et al., 2020) and report results in terms of validation loss.

**Models**. We start from the GaLore Codebase (Zhao et al., 2024a) and train LLaMA models of three sizes: 130M, 350M, and 1B. The models have widths of 768, 1024, and 2048 and depths of 12, 16, and 24. We use the 130M model to explore various ablations as shown in Section 4.1. The MLP hidden dimension of the 130M model is $4$ times the width and the hidden dimension of the 350M and 1B model is $\frac{8}{3}$ times the width. The activation function is SiLU (Elfwing et al., 2018). The architecture uses RoPE positional encodings (Su et al., 2024). Attention heads are always dimension 64. For more architecture details please refer to Zhao et al. (2024a). We train in mixed precision with FP32.

**Algorithms**. We use the standard Pytorch implementation of Adam, and the official GaLore implementation provided by Zhao et al. (2024a). Since Two-Sided SOAP is too memory consuming and is not within our comparison scope, we modify the code provided by Vyas et al. (2024) to apply One-Sided SOAP discussed in Vyas et al. (2024). We use the official Adam-mini implementation provided by Zhang et al. (2024b). For MUON and NS5, we use their implementation provided by Jordan et al. (2024) in their Github records. We implement our COSMOS starting from an older version of Pytorch implementation of AdamW.

**Default hyperparameters.** In all algorithms, we choose first order momentum $\beta_1 = 0.9$ to align with and get a fair comparison with Adam baseline. We choose second order momentum $\beta_2 = 0.98$, which is also a widely used configuration after Liu et al. (2019) mentioned that it provides better training stability than 0.999. We set smoothing term $\epsilon = $ 1e-8 to align with the standard hyperparameter choice. We use the linear learning rate schedule to decay the learning rate to 0. To align with Zhao et al. (2024a), we set the warmup ratio to be 10% and weight decay to be 0.

**Token counts.** For all of our runs we use a sequence length of 1024. For the 130M model, we set the batch size to be 960, and for the 350M and 1B models, we set the batch size to be 2000. We train the 130M and 350M models for 5k steps and train the 1B model for 13k steps. Thus the number of training tokens for the 130M mode $\approx$ 5B, which is beyond the "chinchilla optimal" number of tokens. The numbers of training tokens for the 350M model and 1B model are 10B and 26B respectively, which follow the chinchilla optimal" number of tokens.

### A.1  LEARNING RATE TUNING

To avoid unfair comparisons caused by excessive hyperparameter tuning, for all algorithms we set the learning rate as the only tunable hyperparameter in all the main results in Section 4. The rank $r$ for COSMOS for all main results is fixed at 64.

#### A.1.1  TUNING ON 130M MODEL

For Adam, we tune the learning rate on {2.5e-4, 5e-4, 1e-3, 2e-3, 4e-3, 8e-3}. In our experiments, 2e-3 is the optimal learning rate and 8e-3 diverges. Then for SOAP, we also tune the learning rate on {5e-4, 1e-3, 2e-3, 4e-3}. For Adam-mini, we just use the optimal learning rate of Adam, which is also 2e-3.

For GaLore, We follow the setting in Zhao et al. (2024a), set rank=256, and scale factor $\alpha = 0.25$. According to Zhao et al. (2024a), the learning rate of Galore should be larger than Adam's. They mentioned that Galore is not sensitive to hyperparameter and they use the same learning rate 1e-2 for all size of models after tuning, we simply tune galore in a range near 1e-2, which is {5e-3, 1e-2, 2e-2, 4e-2}. The projection update frequency is 200 for 20k training steps, thus we decrease it to 50 for our 5k training steps.

For the implementation of MUON and COSMOS, the embedding and output layer will use Adam while other parts will use MUON/COSMOS algorithm. To avoid multiple hyperparameter tuning, we fix the learning rate for embedding and output layer to 2e-3, which is the optimal learning rate for Adam, and only tune the learning rate of hidden layers, whose optimizer is MUON/COSMOS.

To be more specific, we tune the learning rate of hidden layers on {1e-4, 2e-4, 5e-4, 1e-3, 2e-3}. It is worth noting that (Liu et al., 2025) suggests the optimal learning rate for MUON should be 0.2–0.4 times that of the Adam learning rate used for the embedding layer (2e-3 in our setting), which exactly falls within the range we searched.

For COSMOS, as we mentioned before, to avoid tuning $\gamma$, we simply set $\gamma$ to be the ratio of the learning rate of hidden layers to the learning rate of the embedding layer (which is fixed at 2e-3). We find that this trick can provide a satisfactory result without extra tuning on $\gamma$. Please note that we find in many extra experiments that this trick isn't the optimal choice for $\gamma$. Tuning $\gamma$ may output a better result.

### A.1.2 TUNING ON 350M MODEL

For Adam and SOAP, we tune the learning rate on {2.5e-4, 5e-4, 1e-3, 2e-3, 4e-3, 8e-3}, which is same as the range in Section A.1.1. For GaLore, we set the rank to be 384, projection update frequency to be 50, and scale factor $\alpha = 0.25$. Then we tune the learning rate of GaLore on {5e-3, 1e-2, 2e-2, 4e-2}, which is also same as what we do in Section A.1.1.

For the implementation of MUON and COSMOS, we still fix the learning rate for embedding and output layer to be 2e-3 and only tune the learning rate of MUON/COSMOS for hidden layers. For MUON and COSMOS, we tune the learning rate on {1e-4, 2e-4, 5e-4, 1e-3, 2e-3} to align with our setting in Section A.1.1. Also for COSMOS, we still set $\gamma$ to be the ratio of the learning rate of hidden layers to the learning rate of the embedding layer (which is fixed at 2e-3).

### A.1.3 TUNING ON 1B MODEL

We do not have enough resources to tune hyperparameters carefully on the 1B model. For Adam, we first try learning rate $\eta = 2e-3$, but an extremely large loss spike occurred in the early stage. Then we decrease $\eta$ to 1e-3 and get the baseline result. For MUON and COSMOS, we still fix the learning rate for embedding and output layer to be 2e-3 and tune their learning rate on {2e-4, 5e-4}. For COSMOS, we still set $\gamma$ to be the ratio of the learning rate of hidden layers to the learning rate of the embedding layer.

### A.1.4 DISCUSSION ON LEARNING RATE TUNING

We are discussing the learning rate used by MUON in their reproducible logs here to demonstrate that our learning rate falls within a reasonable range.

There are two versions of Muon implemented in the reproducible logs of modded nanogpt. In the early versions, the algorithm they used was consistent with what we described in Algorithm 3. This algorithm has been used on both 124M and 1.5B GPT models and achieved SOTA performance.

In this version, they used Adam's baseline learning (3.6e-3) rate as the learning rate for the embedding and output layers on a 124M model, and used 3.6e-4 as the cleaning rate for Muon. In our experiment, since the Adam baseline learning rate we obtained was 2e-3, which is a little different with 3.6e-3, we also use this learning rate as the learning rate for the embedding and output layers. We avoid adjusting the learning rates of the embedding and output layers, as this would result in the tuning of both learning rates for two parts of parameters. Generally speaking, this would yield better results than adjusting only one learning rate, but this effect is not fair compared to the Adam algorithm with only one learning rate. For Muon's learning rate in our experiments, our traversal set {1e-4, 2e-4, 5e-4, 1e-3, 2e-3} is also relatively close to their 3.6e-4.

In the later reproducible logs, they made a simple modification to Muon, but did not mention whether this would improve the effect. This modification is to change the line 7 in Algorithm 3 to be $W_{t+1} \leftarrow W_t - \eta B_t \cdot \sqrt{\frac{m}{n}}$.

Considering that $n$ for different matrix parameters in the same LLM are the same (e.g., 768 in the 124M GPT), this method is just a simple rescale. However, due to the current scale being reduced by $\sqrt{n}$ times, this method generally requires a larger learning rate. For example, in their subsequent logs, they used 0.02 This learning rate is nearly equivalent to using $0.02/\sqrt{768} \approx 7.2e-4$ in the first version, which is also close to our traversal set {1e-4, 2e-4, 5e-4, 1e-3, 2e-3}.

Since MUON did not specify which version would be more advantageous, we used the first version in our experiments, as it provided more reproducible logs. At the time we began experimenting with MUON, the second version was not yet available. For consistency, we therefore continued with the first version, which was also adopted in the subsequent work on MUON scaling (Liu et al., 2025).

### A.2 ABLATION OF $r$ AND $\gamma$ ON 130M MODEL

For the ablation experiments on COSMOS in Section 4.1, we tune discount factor $\gamma$ and rank $r$ together to show COSMOS isn't very sensitive to hyperparameters. We fix the learning rate for embedding and output layers to be $2e-3$, fix the learning rate for COSMOS to be 5e-4, and sweep over the cross product of $r \in \{32, 64, 128\}$ and $\gamma \in \{0.1, 0.25, 0.5, 1\}$. With all these hyperparameters COSMOS outputs comparable results to MUON.

### A.3 ABLATION OF NORMALIZATION

As mentioned in the normalization paragraph in Section 4.1, to exclude the possibility that the better performance of COSMOS than MUON is simply because the normalization function NORM, we modify the normalization method of MUON to be NORM and rerun the experiments for 130M and 350M models. We still tune the learning rate of MUON + NORM on {5e-3, 1e-2, 2e-2, 4e-2}, and present their best performance.

### A.4 PROFILING EXPERIMENTS

We do the profiling experiments on the 1B model. We set the sequence length to 1024, which aligns with our previous settings. We set batch size 10 and accumulation steps 25. Then we record the maximum GPU memory usage and time usage on this setting by using Pytorch API during the entire forward-backward and optimizer update process.

## B  EXPERIMENTS DETAILS ON MODDED-NANOGPT

As discussed in Section 4.3, we directly use Muon's reproducible logs on modded NanoGPT. In the setting of GPT-2 Small (124M), they set the learning rate for embedding layer (optimized with Adam) to be 3.6e-3, and the learning rate for hidden layers (optimized with Muon) to be 3.6e-4. Also they use Warmup-Stable-Decay (WSD) schedule instead of Cosine schedule. Their batch size is 512, sequence length is 1024 and number of iterations is 6200. We also plot the wall-clock time comparison in Figure 6.

In the training of 124M model, we followed their original setting for Muon and only additionally searched $\beta_1$ within $\{0.9, 0.95\}$. For COSMOS, we adopted the same setting as Muon, also searching $\beta_1$ in $\{0.9, 0.95\}$. For $\beta_2$ and $\gamma$ in COSMOS, we set them to $0.95$ and $0.2$ without additional search.

In the training of GPT-2 Medium (350M), they used a very uncommon setting: the learning rate for embedding layer is 0.3, which is very large. But the learning rate for output layer is still 3e-3. They also reduced the momentum for the embedding layer and output layer to $0.8$ – which is also an uncommon choice.

To demonstrate the generality of COSMOS across various settings, our experiments on the 350M model completely follow their setting. We only additionally search $\beta_1$ within $\{0.9, 0.95\}$ for both Muon and COSMOS. However, since this setting is indeed uncommon and subsequent work on Muon scaling (Shah et al., 2025; Liu et al., 2025) did not follow it, we did not adopt this setting in our other experiments (LLaMA on C4 and GPT2 on Wikitext103).

## C  TWO-SIDED COSMOS

For simplicity, we only consider the one-side version of COSMOS in this paper. Like SOAP, COSMOS can be further generalized to a two-sided version. Similar to two-sided SOAP in Vyas et al. (2024), we provide a two-sided variant of COSMOS in Algorithm 4.

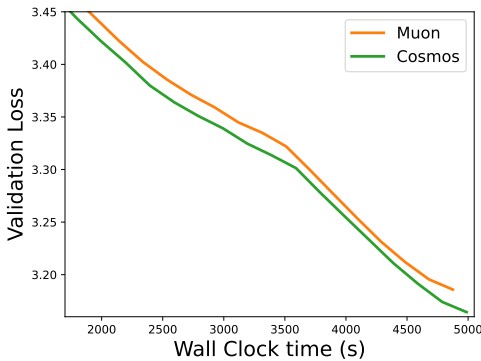

Figure 6: The validation loss vs. wall clock time plot for 350M NanoGPT trained on FineWeb.

---

**Algorithm 4** Two-sided version of COSMOS for a $m \times n$ layer. Per layer, we maintain six matrices: $U \in \mathbb{R}^{n \times r}, O \in \mathbb{R}^{m \times r}, S, R \in \mathbb{R}^{r \times r}, V \in \mathbb{R}^{m \times r}$ and $M \in \mathbb{R}^{m \times n}$.

**input** Learning rate $\eta$, combination weight $\gamma$, projection rank $r \ll n$, momentum parameters $(\beta_1, \beta_2)$, perturbation parameter $\epsilon$. For simplicity, we omit the initialization.

1: **for** $t = 0, \ldots$ **do**
2:      Sample batch $\mathcal{M}_t$
3:      $G_t \leftarrow \nabla_W \phi_{\mathcal{M}_t}(W_t)$
4:      $M_t \leftarrow \beta_1 M_{t-1} + (1 - \beta_1) G_t$
5:      $U_t \leftarrow \text{QR}(\beta_2 U_{t-1} S_{t-1} + (1 - \beta_2) G_t^\top G_t U_{t-1})$
6:      $O_t \leftarrow \text{QR}(\beta_2 R_{t-1} O_{t-1} + (1 - \beta_2) G_t G_t^\top O_{t-1})$
7:      $S_t \leftarrow U_t^\top (\beta_2 U_{t-1} S_{t-1} U_{t-1}^\top + (1 - \beta_2) G_t^\top G_t) U_t$
8:      $R_t \leftarrow O_t^\top (\beta_2 O_{t-1} R_{t-1} O_{t-1}^\top + (1 - \beta_2) G_t G_t^\top) O_t$
9:      $V_t \leftarrow \beta_2 V_{t-1} + (1 - \beta_2)(O_t^\top G_t U_t) \odot (O_t^\top G_t U_t)$
10:      $A_t = O_t \left( \dfrac{O_t^\top M_t U_t / (1 - \beta_1^t)}{\sqrt{(V_t + \epsilon) / (1 - \beta_2^t)}} \right) U_t^\top$
11:      $B_t \leftarrow \text{NORM} \left( \text{NS5} \left( \dfrac{M_t - O_t^\top O_t M_t U_t U_t^\top}{\| M_t - O_t^\top O_t M_t U_t U_t^\top \|_\text{F}} \right) \right)$
12:      $\tilde{G}_t \leftarrow A_t + \gamma \cdot B_t \cdot \sqrt{m}$
13:      $W_{t+1} \leftarrow W_t - \eta \cdot \text{NORM}(\tilde{G}_t)$
14: **end for**

---

# D  ADDITIONAL EXPERIMENTS

This section provides supplementary experiments not presented in the main text.

## D.1  EXPERIMENTS FOR NORMALIZATION ABLATION

We conduct additional experiments on LLaMA-130M and LLaMA-350M to validate that normalization is not the main reason for COSMOS outperforming MUON. As shown in Figure 7, normalization does not make much difference to MUON, while COSMOS consistently outperforms both of them.

## D.2  EXPERIMENTS ON WIKITEXT

This section discuss the details of the experiments on WikiText (Merity et al., 2016) and GPT-2 (Radford et al., 2019). To be more specific, we train GPT2-small(125M) and GPT2-medium (355M) on the Wikitext-103 dataset. We discard learnable position embeddings and use RoPE (Su et al., 2024) as a replacement.

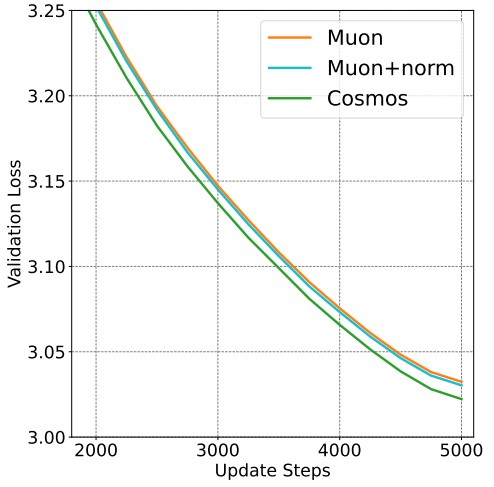 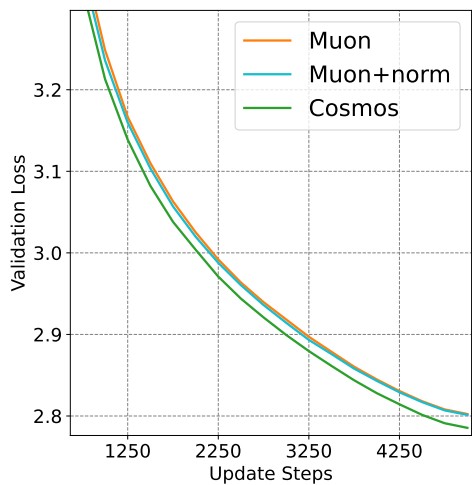

(a) Comparison of COSMOS, MUON and MUON with normalization for LLaMA-130M on C4.

(b) Comparison of COSMOS, MUON and MUON with normalization for LLaMA-350M on C4.

Figure 7: Comparison of COSMOS, MUON, and MUON with normalization on LLaMA-130M and LLaMA-350M for C4.

For GPT2-small, we tune the learning rate of Adam on {5e-4, 1e-3, 2e-3, 4e-3, 8e-3}, and find 4e-3 is the optimal learning rate for Adam. Then in MUON/COSMOS, we use learning rate 4e-3 for the embedding layer and 4e-4 for MUON/COSMOS.

Similarly, for GPT2-medium, we tune the learning rate of Adam on {5e-4, 1e-3, 2e-3, 4e-3, 8e-3}, and find 2e-3 is the optimal learning rate for Adam. Then in MUON/COSMOS, we use learning rate 2e-3 for the embedding layer and 5e-4 for MUON/COSMOS.

For COSMOS, $\gamma$ is still set to be the ratio of the hidden layer learning rate to the embedding layer learning rate in both models.

We set the sequence length to be 1024, and the batch size is also 1024. We train both models for 5k steps, which means the models are trained on 5B tokens. For such many training tokens on Wikitext-103, overfitting will occur and validation loss will start to increase after training for a certain number of steps. Therefore, we use the training loss as the metric for comparison.

The results for GPT2-small and GPT2-medium are provided in Figures 5a and 5b, respectively. We observe that COSMOS consistently outperforms MUON, showing that it does not overfit any particular model or dataset.

### D.3 SMALLER LEARNING RATE FOR MUON/COSMOS ON LLAMA-1B

As discussed in section A.1.3, we tune the learning rate for MUON/COSMOS on {2e-4, 5e-4}. We find 5e-4 is better and present its corresponding results in the main text. Here we present the result for 2e-4 in Figure 8, where COSMOS still outperforms MUON.

### D.4 WALL-CLOCK TIME PLOT FOR LLAMA-1B

Based on the throughput we calculate in Table 7, we rescale the X-axis of Figure 2b to be wall-clock time and present the result in Figure 9.

### D.5 EXPERIMENTS ON SHORTER SEQUENCES

To validate the correctness of our implementation, we compare GaLore with COSMOS and Adam on 256 sequence length as adopted in Zhao et al. (2024a). As shown in Figure 10, GaLore and Adam

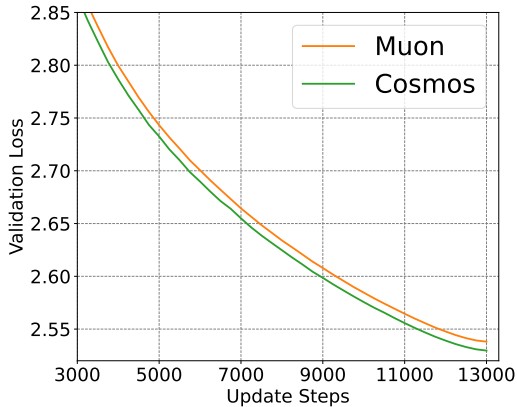

Figure 8: LLaMA-1B trained on C4 dataset with learning rate 2e-4 for MUON/COSMOS. COSMOS still outperforms MUON.

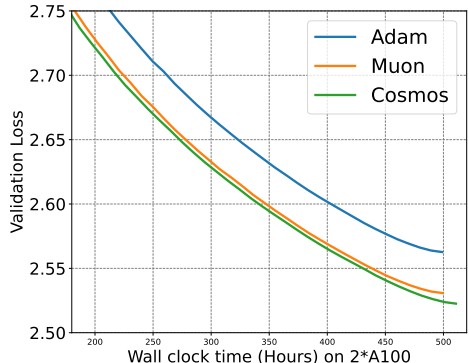

Figure 9: Wall-Clock time plot for our training on LLaMA-1B.

are more comparable in this shorter sequence setting, suggesting our implementation is correct and the degradation of GaLore shown in Figure 1 and Table 2 is mainly due to long sequence length.

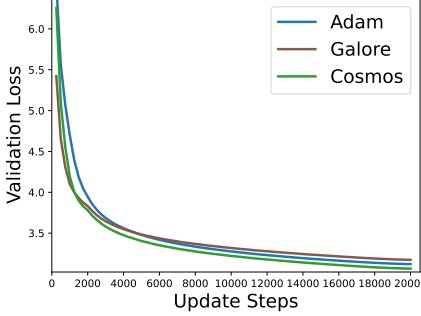

Figure 10: Comparison of COSMOS, Adam, and GaLore on 256 sequence length. The performance of GaLore on shorter sequences does not deteriorate as for long sequences, validating the correctness of our implementation.

## E LLM USAGE

In preparing this paper, large language models (LLMs) such as ChatGPT were used only for light editing purposes, including minor grammar checking and sentence polishing. No part of the research ideation, methodology design, experimental execution, or analysis was conducted with the assistance of LLMs.

