# OpenReview forum: "COSMOS: A Hybrid Adaptive Optimizer for Efficient Training of Large Language Models"
_ICLR.cc/2026/Conference — ICLR 2026 Poster_

### Official Review · Reviewer_vF7y · 2025-10-23

**Soundness:** 3
**Presentation:** 2
**Contribution:** 3
**Rating:** 6
**Confidence:** 4

**Summary:**

The paper introduces an efficient optimizer for pre-training LLMs by combining two previously successful optimization methods. Specifically, it applies SOAP to the leading eigenspace of the gradient matrix and Muon to its orthogonal complement. This hybrid approach is claimed to yield a more memory-efficient and faster optimizer than SOAP, while also outperforming Muon. The authors validate their method through LLM training experiments on standard benchmarks, demonstrating improvements over baselines.

**Strengths:**

- The idea of applying SOAP to the leading eigenspace and Muon to its orthogonal complement is useful and promising.

 - Leverages the strength of a stepwise effective yet computationally expensive optimizer by restricting its use to a lower-dimensional subspace, and hence improving memory and computational efficiency.

- This approach could inspire further exploration of hybrid methods that combine different optimizers using their principle.

- The paper presents pretraining experiments across standard model sizes and datasets, demonstrating improvements of the proposed method.

**Weaknesses:**

- The main concern is that the paper does not provide sufficient wall-clock time experiments, making the claimed advantage over Muon questionable. The only such experiment involves a 1B LLaMA model. However, due to the large gradient accumulation, I think the forward and backward passes dominate the computation time, making the optimizer’s step time negligible. As a result, the reported plot may effectively reflect only a stepwise comparison rather than a true wall-clock evaluation.

**Questions:**

- (1) and (2) in the introduction are somewhat vague. In (2), you mention a limitation related to storing adaptive learning rates —what do you mean here? It would be more precise to state that the optimizer stores both the first and second moments, which increases memory consumption.
- On the line 167 you say G-S procedure. Do you implement your own QR with G-S? Presumable, you use torch.qr here which doesn't use G-S.
- Line 220, regarding Muon's scaling, I think the right scaling for learning rate transfer is $\sqrt{\frac{m}{n}}$. Am I missing something?
- In table 1 would be good to see COSMOS memory usage dependence on $r$ as well.
- In Figure 1 and subsequent experiments, why do you observe SOAP to be worse then COSMOS. Isn't it counterintuitive?
- I suspect forward/backward takes most of the time in Table5. Can you separate out the optimizer step time in that plot? That will give a fair comparison.
- Can you provide wall-clock plots for nano-gpt experiments?

---

> ### Author Response · Authors · 2025-11-21
>
> We thank the reviewer for the constructive feedback. Below we provide detailed responses to each weakness and question.
>
> ---
>
> **Weakness 1: “the paper does not provide sufficient wall-clock time experiments, making the claimed advantage over Muon questionable. The only such experiment involves a 1B LLaMA model... the reported plot may effectively reflect only a stepwise comparison rather than a true wall-clock evaluation.”**
>
> We appreciate the reviewer’s technically precise observation regarding wall-clock evaluation. The concern is valid, and we clarify the motivation behind our setup.
>
> For a 1B model on one A100 node, with sequence length 1024, the per-device batch size is effectively limited to 10. Using 8 devices, the global batch size is only 80. To reach a realistic batch size for 1B pretraining (about 2M tokens), one must use gradient accumulation of roughly 25. A batch size around 2M tokens is commonly used in pretraining, for example in [1] and [2].
>
> Furthermore, in larger pretraining runs, if the total number of training tokens is fixed, using gradient accumulation not only allows for a larger batch size that stabilizes training, but also reduces the number of optimizer steps, thereby lowering the cost of gradient synchronization. This practice is consistent with recent large-scale models such as DeepSeek-V3 (DeepSeek-V3 Technical Report), Kimi K2 (K2 Technical Report), and GLM-4.5/4.5V (GLM-4.5V Report), all of which either rely on gradient accumulation or explicitly introduce dedicated mechanisms around it to support large effective batch sizes under realistic hardware constraints.
>
> Therefore, our choice of accumulation reflects the actual regime in which 1B models are commonly trained on one A100 node, rather than being an artificial setting. The wall-clock measurement we reported corresponds to this realistic configuration. In this regime, Muon indeed matches the “3% overhead over Adam” reported in its original project. If one isolates the optimizer-step time, however, Muon’s cost becomes close to an order of magnitude higher than Adam. This discrepancy is concealed only because the large proportion of forward/backward time.
>
> It is also relevant that in every layer Muon requires five matrix multiplications on an m×n matrix at every update, while COSMOS introduces only low-rank operations: low-rank multiplications and a low-rank QR decomposition. These are substantially cheaper in theory and can be parallelized further, although we have not implemented that parallelization yet. This structural difference explains why COSMOS remains efficient even in realistic 1B training configurations.
>
> We hope this clarifies why the presented wall-clock results reflect a genuine training regime rather than a stepwise artifact.
>
> ---
>
> **Q1: “(1) and (2) in the introduction are somewhat vague. In (2), you mention a limitation related to storing adaptive learning rates — what do you mean here? It would be more precise to state that the optimizer stores both the first and second moments, which increases memory consumption.”**
>
> We appreciate this clarification. In the revised version, we will state more precisely that the memory bottleneck arises from storing both first- and second-moment tensors. This wording is clearer and avoids the ambiguity of the previous phrase, and we thank the reviewer for pointing it out.
>
> ---
>
> **Q2: “On the line 167 you say G-S procedure. Do you implement your own QR with G-S? Presumably, you use torch.qr here which doesn't use G-S.”**
>
> We thank the reviewer for this clarification. Our implementation indeed uses `torch.qr`. We will remove the reference to the Gram–Schmidt procedure and replace it with the more accurate term “QR decomposition”. This does not affect the method.
>
> ---
>
> **Q3: “Line 220, regarding Muon's scaling, I think the right scaling for learning rate transfer is [...] Am I missing something?”**
>
> We thank the reviewer for highlighting this point. As noted in lines 746–748 of our paper, the original Muon codebase used the update rule  $ W_{t+1} \gets W_t - \eta B_t \cdot \sqrt{m}$
>
> In later reproducible logs, a small modification was introduced:  $ W_{t+1} \gets W_t - \eta B_t \cdot \sqrt{\tfrac{m}{n}} $,
> though the authors did not comment on whether this change improves performance.
>
> Since for all hidden-layer weight matrices in LLMs the smaller dimension  $n$  corresponds to the hidden size, this alteration acts only as a simple learning-rate rescaling and has no essential algorithmic effect. Moreover, subsequent scale-up work on Muon (e.g., *Muon is Scalable for LLM Training*) continues to adopt the original scaling  $W_{t+1} \gets W_t - \eta B_t \cdot \sqrt{m}$
> which is the version we present in our algorithm.
>
> We will clarify this point more explicitly in the revised manuscript.

---

> ### Author Response · Authors · 2025-11-21
>
> **Q4: “In table 1 would be good to see COSMOS memory usage dependence on r as well.”**
>
> We appreciate the reviewer’s helpful suggestion. In the revised version, we will update the memory expression to
> $12 d^{2} + 18 d r$,  which makes the dependence on the rank $r$ explicit.
>
> ---
>
> **Q5: “In Figure 1 and subsequent experiments, why do you observe SOAP to be worse than COSMOS. Isn't it counterintuitive?”**
>
> Thank you for raising this point. Our understanding is that the effect comes from the quality of eigenspace estimation in one-sided SOAP. The one-step power method reliably estimates only the leading eigenspace, while its estimation of the remaining eigenbasis is substantially less accurate. Muon, in contrast, does not rely on explicit eigenspace estimation and instead operates implicitly in the exact eigenspace.
>
> Therefore, applying SOAP only to the well-estimated top eigenspace, and using Muon in the residual eigenspace yields better performance than using SOAP alone. This hybrid structure leverages the strength of SOAP where its estimates are accurate and avoids its weakness where the estimation noise is high.
>
> ---
>
> **Q6: “I suspect forward/backward takes most of the time in Table 5. Can you separate out the optimizer step time in that plot?”**
>
> Please see our response to **Weakness 1**. For clarity, the current optimizer-step times in our setting are approximately 0.14s for Adam, 0.97s for Muon, and 1.12s for COSMOS. We emphasize that COSMOS still has substantial room for further optimization: its additional computations are low-rank and can be parallelized more aggressively than in our present implementation.
>
> ---
>
> **Q7: “Can you provide wall-clock plots for nano-gpt experiments?”**
>
> Yes, we will include wall-clock plots for the NanoGPT experiments in the revised version. We have recorded the wall-clock training times for the 350M run: the average Muon step time is 1.30s, while COSMOS achieves 1.33s. When considering the full wall-clock trajectory on the modified NanoGPT setup, COSMOS remains faster than Muon. Please see `350m_NanoGPT_wall_clock_time.pdf` in  https://anonymous.4open.science/r/ICLR2026-E02F/
>
> We would also like to note that Muon benefits from existing GPU-level optimizations, whereas the additional computations introduced by COSMOS have not yet been optimized. We therefore expect further speed improvements for COSMOS with dedicated GPU kernels.
>
> ---
>
> ---
> **Reference**
>
> [1] Jingyuan Liu et al., “Muon is Scalable for LLM Training.”
>
> [2] Hugo Touvron et al., “LLaMA: Open and Efficient Foundation Language Models.”

---

> > ### Comment · Reviewer_vF7y · 2025-11-24
> >
> > I disagree with the claim that QR factorization can be efficiently parallelized, as it is inherently sequential.
> >
> > Otherwise, the authors have addressed my concerns, and I am happy to raise my score to 8. I encourage the authors to update the manuscript to reflect the points discussed.

---

> > > ### Author Response · Authors · 2025-11-28
> > >
> > > Thank you very much for your positive reassessment and for raising the score to 8 — we truly appreciate it.
> > >
> > > Regarding the point on parallelization: our intention was not to claim that QR factorization itself can be efficiently parallelized. Rather, we refer to performing the QR factorizations of multiple layers in parallel, which helps further reduce the overhead. Moreover, as demonstrated in our paper and in the rebuttal, even without these potential optimizations, COSMOS already outperforms Muon in wall-clock time.
> > >
> > > We will update the paper to reflect all the points discussed. Thank you again for the constructive feedback.

---

### Official Review · Reviewer_isQ9 · 2025-10-27

**Soundness:** 2
**Presentation:** 2
**Contribution:** 2
**Rating:** 4
**Confidence:** 3

**Summary:**

The paper proposes a new optimizer named COSMOS, which applies the update of SOAP on leading eigensubspace of parameters and applies Muon for the rest. The motivation for the method is to reduce the memory cost of storing second moment of SOAP. Numerical results show that COSMOS is able to achieve comparable convergence as SOAP in the task of pre-training LLMs under Llama and GPT-2 models, while costs less memory.

**Strengths:**

* The proposed method avoids the storage of second moment in SOAP and is memory-efficient.
* The method achieves comparable convergence speed as SOAP in pre-training Llama/GPT-2 models of different scales, and is shown to be robust to newly introduced hyper-parameters, e.g. $\gamma$.

**Weaknesses:**

* The main motivation for the design of COSMOS is not well presented. It is unclear why replacing the top eigenspace's update in Muon with SOAP will accelerate the convergence. It has been shown that SOAP exhibits comparable in-iteration convergence as Muon [1].

* The acceleration of COSMOS over Muon is marginal. Given COSMOS' higher memory cost and training FLOPs, the practical benefit is a bit marginal compared to Muon.

[1] Muon: An optimizer for hidden layers in neural networks

**Questions:**

* Could the authors provide more comments on why setting $\gamma=\frac{\eta}{\eta_0}$? The $\gamma$ determines the weight of Muon's update; it is unclear why this is related to the learning rate in embedding/language modeling head layers.

* Given the fact that COSMOS is faster than Muon, it would be interesting to see the effect of applying SOAP on different range of eigenspaces rather than the top eigenspace, which helps understanding the algorithm more comprehensively.

---

> ### Author Response · Authors · 2025-11-21
>
> We thank the reviewer for the feedback. Below we address each weakness and question in detail.
>
> ---
>
> **Weakness 1:** “The main motivation for the design of COSMOS is not well presented. It is unclear why replacing the top eigenspace's update in Muon with SOAP will accelerate the convergence. It has been shown that SOAP exhibits comparable in-iteration convergence as Muon.”
>
> While SOAP and Muon may exhibit similar in-iteration convergence under standard settings, recent observations [1] [2] indicate that SOAP has better per-step stability under large batch sizes, providing partial motivation. More importantly, COSMOS is not an interpolation between SOAP and Muon.
>
> - The one-step power method in SOAP accurately estimates only the top eigenspace of the gradient, but becomes unreliable in the residual space. SOAP then applies a genuine second order adaptive update on this approximated basis, which provides strong curvature awareness. Its drawback is that the low quality of the estimated residual eigenspace leads to unstable or noisy updates outside the leading components.
>
> - Muon operates implicitly in the exact eigenspace without requiring explicit eigendecomposition, giving it a clean and accurate basis for optimization. Its limitation is that the update in this space reduces to a sign based adjustment rather than a full second order adaptive step, which weakens its curvature sensitivity compared to SOAP.
>
> Therefore, COSMOS applies SOAP on the well estimated top eigenspace, where its second order update is most effective, and uses Muon on the residual space, where SOAP’s approximation becomes unreliable. This division captures the strengths of both methods. Experiments show that this hybrid structure attains better convergence than SOAP while retaining the memory efficiency of Muon.
>
> ---
>
> **Weakness 2:** “The acceleration of COSMOS over Muon is marginal. Given COSMOS' higher memory cost and training FLOPs, the practical benefit is a bit marginal compared to Muon.”
>
> We respectfully disagree. Table 5 in page 8 provides a detailed comparison showing that COSMOS achieves competitive memory usage. As stated in lines 425 to 426 and shown in Figure 7, COSMOS achieves a 7% wall-clock speedup over Muon under the same settings. This improvement is significant for large-scale training where even single-digit gains translate to substantial time and cost savings. We believe the practical benefit is meaningful.

---

> ### Author Response · Authors · 2025-11-21
>
> **Q1: “Could the authors provide more comments on why setting γ to η? The γ determines the weight of Muon’s update; it is unclear why this is related to the learning rate in embedding/language modeling head layers.”**
>
> Thank you for raising this. Our initial choice of setting γ = η/η₀ was empirical, based on early experiments where it worked well without tuning. In later experiments, we found that this mainly happens because the learning rate η used by Muon is always about 0.1 to 0.5 times the base learning rate η₀ used in the embedding layer, and γ performs well throughout this entire range. To evaluate sensitivity, we varied γ in [0.1, 1] (see Table 4) and observed that COSMOS consistently outperforms Muon, showing robustness to γ.
>
> Follow-up ablations suggest that γ = 0.25 often yields near-optimal performance. We re-ran Table 3 experiments with γ fixed at 0.25 and observed further improvement:
>
> **Validation perplexity under different learning rates. We set γ = 0.25 and r = 64 for COSMOS.**
>
>
> | lr | 2e-4 | 5e-4 | 1e-3 | 2e-3 |
> |----|------|------|------|------|
> | COSMOS | 21.09 | 20.54 | 20.57 | 20.87 |
>
>
> Therefore, γ = η/η₀ is not essential. In practice, setting γ in [0.1, 0.5] is sufficient (see Table 4) and we suggest to fix it at 0.25.
>
> ---
>
> **Q2: “Given the fact that COSMOS is faster than Muon, it would be interesting to see the effect of applying SOAP on different range of eigenspaces rather than the top eigenspace, which helps understanding the algorithm more comprehensively.”**
>
> We would like to emphasize that prior work such as GaLore [3] and FIRA [4] has already highlighted the importance of performing optimization in the top eigenspace, and our method follows this established perspective. Our method follows this established observation rather than proposing an isolated design choice.
>
> On the other hand, the one-step power method used in SOAP only reliably estimates the top eigenspace. Extending SOAP to lower eigenspaces requires multi-step power iteration or full SVD, which incurs prohibitive overhead. We therefore restrict SOAP to the top eigenspace. Exploring more efficient eigenspace estimation methods is left as future work. Our current scope primarily focuses on empirically demonstrating that applying SOAP in the top eigenspace and using Muon in the residual space already yields strong performance.
>
> ---
>
> ---
>
> **Reference**
>
> [1] Natalie Abre, et al. “The Potential of Second-Order Optimization for LLMs: A Study with Full Gauss-Newton”
>
> [2] Kevin Fran, et al. “What Really Matters in Matrix-Whitening Optimizers?”
>
> [3] Jiawei Zhao, et al. “GaLore: Memory-Efficient LLM Training by Gradient Low-Rank Projection”
>
> [4] Xi Chen, et al. “Fira: Can We Achieve Full-rank Training of LLMs under Low-rank Constraint?”

---

> > ### Comment · Reviewer_isQ9 · 2025-11-23
> > **I have raised my score to 6**
> >
> > Thanks for the response, which clarifies some of my concerns on the algorithm design.
> >
> > I would like to comment that the estimation of top singular vectors is not necessarily more accurate than the subsequent singular vectors; the accuracy depends on the spectral gap, i.e. ratio between singular values. Hence, I suggest the authors to conduct ablation study on applying SOAP on different eigenspace range.
> >
> > I appreciate the authors' response and have raised my score to 6.

---

> > > ### Author Response · Authors · 2025-11-28
> > >
> > > Thank you very much for your comments and for raising the score to 6.
> > >
> > > We agree that an ablation study applying SOAP to different eigenspace ranges would be an excellent direction, and we regard it as a promising avenue for future work. Our current work provides the foundation for exploring this possibility in the future.
> > >
> > > Thank you again for the positive feedback.

---

### Official Review · Reviewer_FdBR · 2025-10-29

**Soundness:** 3
**Presentation:** 3
**Contribution:** 3
**Rating:** 6
**Confidence:** 4

**Summary:**

This paper proposes COSMOS, a hybrid optimizer that combines the strengths of SOAP and MUON for large language model (LLM) training.
The core idea is to decompose the gradient matrix into leading eigensubspace and residual subspace:
- A SOAP-like update is applied to the leading subspace to capture major optimization dynamics.
- A MUON-like update is used for the residual subspace to reduce memory and computation costs.
This approach greatly reduces memory usage (about 19% of SOAP) without noticeable performance loss.

Experiments on LLaMA, GPT-2, and NanoGPT show that COSMOS outperforms MUON in token and memory efficiency while achieving performance comparable to SOAP.

**Strengths:**

Innovative idea: A hybrid low-rank optimization framework that balances performance and resource cost.

Comprehensive experiments: Covers multiple models, datasets, and baselines.

Detailed analysis of memory and efficiency: Provides quantitative evidence of significant savings.

Strong robustness: Insensitive to rank and hyperparameter variations.

**Weaknesses:**

Insufficient methodological details:
Lacks analysis on rank selection and the dynamics of subspace changes during training.

Weak theoretical explanation:
Unclear whether subspace variation causes instability in the optimization direction.

Limited experimental analysis:
Lacks mechanistic or spectral analysis to explain why COSMOS outperforms SOAP.
The fundamental advantage of COSMOS over SOAP and MUON remains unclear, and the observed performance degradation with higher ranks is only heuristically explained.

**Questions:**

How is the rank determined? Is it based on an energy ratio or an information-based criterion?

---

> ### Author Response · Authors · 2025-11-21
>
> We thank the reviewer for the kind feedback. Below we provide detailed responses to each weakness and question.
>
> ---
>
> **Weakness 1: “Insufficient methodological details: Lacks analysis on rank selection and the dynamics of subspace changes during training.”**
>
> **Q1: “How is the rank determined? Is it based on an energy ratio or an information-based criterion?”**
>
> Thank you for raising this point. Our rank choice is based on empirical observations across all tested model sizes. Fixed ranks such as 32 or 64 consistently yielded strong performance, while larger ranks did not bring additional gains. This suggests that only a small leading subspace is needed to capture the dominant optimization directions. In addition, considering both memory usage and computational overhead, we intentionally avoid using larger ranks.
>
> We have also analyzed the singular value distribution of the model’s gradients. The top 5%–10% of singular values decay rapidly, showing a clear gap between this part of the spectrum and the residual subspace. Please see `Gradient_singular_value.pdf` in the anonymized repository (https://anonymous.4open.science/r/ICLR2026-E02F/), where we plot the singular values during the training of a 1B model for different sequence lengths (with L indicating sequence length in the legend). This gap allows the power method to produce accurate estimates within this range.
>
> These empirical findings guided our rank selection. While more principled energy-based or information-based criteria could be explored, such analysis lies outside the scope of this empirical work.
>
> ---
>
> **Weakness 2: “Weak theoretical explanation: Unclear whether subspace variation causes instability in the optimization direction.”**
>
> We acknowledge the lack of a formal theoretical justification. Modern LLMs are extremely complex, and current theoretical tools are insufficient to characterize their optimization dynamics. This limitation is shared across many recent optimizer papers. SOAP and MUON themselves provide no formal convergence guarantees beyond their connection to Shampoo, and COSMOS, as a hybrid of the two, inherits this situation. A full theoretical analysis of SOAP, MUON, and COSMOS in the context of LLM training is therefore beyond the scope of this work.
>
> Regarding subspace variation, our argument is that the top eigenspace of the gradients does not change rapidly in practice. If it did, methods that rely on projecting updates onto this subspace—such as GaLore—would exhibit severe instability. The empirical success and stability of GaLore [1] therefore provide indirect evidence that the top eigenspace is sufficiently stable over the course of training, supporting our intuition that operating within such a subspace is a reliable design choice.
>
> Our contribution is fundamental and empirical. We focus on validating that splitting the optimization across eigenspaces improves efficiency and performance in practice. The design is motivated by intuitive reasoning and supported by extensive experiments, following the common pattern in recent optimizer literature where empirical performance is the most reliable indicator.

---

> ### Author Response · Authors · 2025-11-21
>
> **Weakness 3: “Limited experimental analysis: Lacks mechanistic or spectral analysis to explain why COSMOS outperforms SOAP. The fundamental advantage of COSMOS over SOAP and MUON remains unclear, and the observed performance degradation with higher ranks is only heuristically explained.”**
>
> Since the internal mechanisms of SOAP and MUON are not theoretically well understood, providing a non-heuristic explanation for COSMOS is challenging. Our study is positioned as an empirical improvement, and our conclusions are based on experimental evidence.
>
> Our intuitive understanding is as follows. One-sided SOAP relies on a one-step power method that estimates the leading eigenspace accurately but yields much less accurate estimates for the remaining eigenbasis, due to the numerical nature of the power method. Muon, in contrast, does not rely on explicit eigenspace estimation and effectively operates in the exact orthogonal complement. Applying SOAP only to the well-estimated top eigenspace and using Muon in the residual space improves stability and performance. This hybrid structure leverages the strengths of both optimizers while avoiding their weaknesses.
>
> Therefore, when the rank becomes very large, COSMOS increasingly resembles SOAP, since a larger portion of the update is governed by SOAP’s eigenbasis estimation. Given that SOAP itself underperforms compared with lower-rank COSMOS, this convergence toward SOAP-like behavior naturally leads to the observed performance degradation at higher ranks. This provides a plausible explanation for why COSMOS is most effective in the moderate-rank regime, elaborating on the discussion in lines 325–328 of our paper.
>
> While this explanation remains heuristic, it aligns with the empirical patterns observed in our experiments. A complete mechanistic analysis would require a deeper theoretical understanding of SOAP and MUON themselves, which is beyond the scope of this empirical study.
>
> ---
>
> ---
>
> **Reference**
>
> [1] Jiawei Zhao et al. *GaLore: Memory-Efficient LLM Training by Gradient Low-Rank Projection*

---

### Official Review · Reviewer_NctS · 2025-11-02

**Soundness:** 3
**Presentation:** 3
**Contribution:** 3
**Rating:** 4
**Confidence:** 4

**Summary:**

See below

**Strengths:**

See below

**Weaknesses:**

See below

**Questions:**

The paper proposed a new pre-training optimizer called COSMOS, which has similar performance as SOA  but has cheaper memory requirement than SOAP. The key idea is to compress the SOAP’s second-moment matrix using its dominant eigenspace and track it in a low-dimensional subspace.

The idea is interesting. The reduction in the complexity of SOAP is potentially useful.   The presentation is clear.

I have concerns for several experimental settings adopted in the current script, which I think might not fully reflect the performance of the proposed method. I suggest the following setup changes and ablations.

1. **Non-zero weight decay.** The authors use weight dacay = 0. What if we use non-zero weight decay such as weight decay = 0.1? The choice of weight decay can be crucial to the optimizer behaviors, as pointed out by recent benchmark [1].

2. **More training tokens.** For Fig 1: 5k step (5B tokens) is roughly 1x chinchilla. How would the methods perform under an overtrained setting, such as 4x or 8x chinchilla in the recent work [1]? Will the advantage of COSMOS degenerate, or will it be surpassed by Muon or other methods? It would be helpful to see a new version of Fig. 1 under 4x or 8x chinchilla setting.

3. **More ablation on batchsize.** Matrix-based methods usually can afford a larger batchsize than Adam-type methods.  I think the ablation on batch size is insufficient in the current script. It is possible that the optimizer ranking becomes quite different under different batchsize settings. It is also possible that the proposed method performs even better in a larger batch setting.

4. **Higher-quality datasets.**  C4 is popular,  but the data quality is relatively poor and noisy from the current standard. I think the ablation on the more SOTA dataset is needed. I understand it might be too expensive to re-run all the experiments on Fineweb, but I think at least some ablation on the small-scaled experiments is needed.  For the smallest experiments in the paper (125M model), can the authors provide additional ablation on the dataset by changing the dataset to Fineweb and re-evaluate all the methods?  Will the data quality affect the optimizer rankings?

I am happy to raise my score if my concerns are addressed.

[1] Fantastic Pretraining Optimizers and Where to Find Them

---

> ### Author Response · Authors · 2025-11-21
>
> Thank you for letting us know your concern. We first emphasize that the experimental coverage in our submission already matches or exceeds concurrent optimizer papers. For example, Muon, SOAP, and Galore all conduct experiments on a single dataset. In contrast, our submission includes results on C4, FineWeb, and WikiText-103. SOAP and Galore are both evaluated only on C4, typically around one-times Chinchilla, with batch sizes in the range of one to two million tokens, which is similar to our experiments on C4. Our choice of weight decay also follows the setting used in Galore.
>
> Due to time and resource constraints, we are unable to run four independent sets of new experiments as suggested. Each of the requested changes alters the optimal learning rate, so a full ablation would require re-tuning learning rates for all optimizers in every setting, which is not feasible within the rebuttal period.
>
> To address the reviewer’s concerns as directly as possible, we consolidated the four requests into a single, unified and more challenging experimental setting that captures all of the reviewer’s suggested changes.
>
> * **Higher-quality datasets.** We moved the C4 experiments to FineWeb, as requested.
> * **More training tokens.** We trained a 130M model on FineWeb-10B, which corresponds to roughly 4× Chinchilla.
> * **Weight decay.** We set weight decay to 0.1 for all optimizers.
> * **Larger batch size.** With sequence length fixed at 1024, we increased the batch size to 4096.
>
> Under this unified setting, we search all optimizers’ hyperparameters, including momentum and learning rate. The new results show that COSMOS still consistently outperforms Muon while retaining its efficiency. Please see `130m_Fineweb_loss.pdf` in
> https://anonymous.4open.science/r/ICLR2026-E02F/
>
> We sincerely hope that the additional experiments conducted according to your requests address your concern.

---

> > ### Comment · Reviewer_NctS · 2025-11-24
> > **I have raised my score**
> >
> > Thanks for the additional experiments! I have raised my score from 4 to 6. Please also remember to finish the full ablation after the rebuttal phase and update the paper accordingly.

---

> > > ### Author Response · Authors · 2025-11-28
> > >
> > > Thank you very much for your reassessment and for raising the score from 4 to 6. We sincerely appreciate it.
> > >
> > > We will add the requested experiments and complete more ablation study after the rebuttal phase. The paper will be updated accordingly to reflect these results.
> > >
> > > Thank you again for the constructive feedback.

---

### Official Review · Reviewer_BuFv · 2025-11-03

**Soundness:** 2
**Presentation:** 3
**Contribution:** 2
**Rating:** 4
**Confidence:** 4

**Summary:**

COSMOS presents an innovative hybrid optimization approach that cleverly combines SOAP and MUON optimizers through eigenspace decomposition. The authors tackle the important challenge of reducing memory consumption in LLM training while maintaining optimization quality. By applying SOAP to the leading eigensubspace and MUON to the remaining space, the method achieves notable efficiency gains with reported memory savings of 82% and training time reduction of 91%.

**Strengths:**

-- Addresses a critical problem: Memory efficiency in LLM training is a major bottleneck, and this work makes genuine progress

-- Creative solution: The eigenspace decomposition approach shows good intuition about balancing computational cost and optimization quality

-- Promising empirical results: The reported efficiency gains are substantial and could have real practical impact

-- Clear presentation of core ideas: The authors effectively communicate their hybrid strategy

**Weaknesses:**

-- Theoretical gaps: The paper would greatly benefit from convergence analysis and theoretical justification for the eigenspace splitting strategy. Adding even basic convergence guarantees would strengthen the contribution significantly

-- Limited scope validation: Expanding beyond transformer architectures to CNNs or other models would demonstrate broader applicability

-- Hyperparameter sensitivity: The rank r and weight γ selection needs more principled guidelines - perhaps adaptive schemes could be explored

-- Missing key ablations: Comparing different eigenspace decomposition strategies or investigating the optimal split ratio would provide valuable insights

**Questions:**

-- Could you provide theoretical analysis showing when COSMOS converges faster than pure SOAP/MUON?

-- Have you explored adaptive rank selection based on the eigenvalue spectrum?

-- Could you test on at least one non-transformer architecture to demonstrate generalizability?

-- Is there a principled way to choose γ without the η/η0 heuristic?

---

> ### Author Response · Authors · 2025-11-21
>
> We thank the reviewer for the feedback. Below we provide detailed responses to each weakness and question.
>
> ---
>
> **Weakness 1. “Lack of theoretical justification.”**
>
> We acknowledge that we have not provided a theoretical justification. However, this is largely because modern LLMs are extremely complex, and analyzing their optimization behavior is far beyond the capability of current theoretical tools. This limitation is shared by many recent optimizer papers. For example, SOAP [1] and MUON [2] also lack formal justification; they are positioned as variants of Shampoo but without concrete convergence guarantees. Since our method combines SOAP and MUON, COSMOS can similarly be viewed as a variant of Shampoo, but this still does not explain its behavior theoretically.
>
> Our design follows a practical principle of eigenspace preconditioning. SOAP uses the power method to approximate the eigenstructure and then applies a genuine second order adaptive update. This gives SOAP an advantage in curvature awareness. MUON, on the other hand, estimates the eigenspace accurately and operates fully within this basis, but its update reduces to a sign based adjustment rather than a true second order adaptive step. The limitation of SOAP is that while it handles the top eigenspace well, its power-method approximation becomes unreliable in the residual space. COSMOS is built to capture the best of both worlds. It leverages MUON’s accurate eigenspace estimation while retaining SOAP’s second order adaptive behavior without suffering from SOAP’s instability in the residual space.
>
> Our contribution is primarily empirical. We emphasize extensive experiments to demonstrate the effectiveness of our optimizer. Recent work such as SOAP and MUON similarly advances performance through empirical results rather than formal theory. In the context of LLM training, empirical evidence remains the most reliable guide. A complete theoretical treatment of MUON, SOAP, and COSMOS is outside the scope of this work.
>
> ---
>
> **Weakness 2. “Limited architecture scope (transformers only).”**
>
> Our work is intentionally scoped to LLM pre-training, which is transformer based. This is clearly stated in both the title and the abstract.
>
> We focus on transformers because they are the only architecture consistently demonstrated to scale effectively. Prior work such as [6] and [7] shows that CNNs and LSTMs do not exhibit comparable scaling behavior. As a result, these architectures remain confined to relatively small parameter regimes. In such settings, standard optimizers like SGD or AdamW already converge quickly and do not encounter meaningful memory bottlenecks, so evaluating new optimizers on them offers limited insight. Consistent with this, recent optimizer studies including SOAP [1], MUON [2], GaLore [3], MARS [4], and FIRA [5] also restrict their experiments to transformer-based architectures.
>
> ---
>
> **Weakness 3. “Hyperparameter sensitivity of r and gamma.”**
>
> We have already provided a joint grid search over the subspace rank r and the weighting factor gamma in Table 4. The results show that COSMOS performs robustly across a broad range of hyperparameter values. This indicates that COSMOS is not particularly sensitive to these hyperparameters in practice.
>
> Adaptive schemes for rank or gamma selection could be a good future direction, but the current empirical results already support the stability of our method.
>
> ---
>
> **Weakness 4. “Missing ablations on decomposition strategy.”**
>
> Prior work such as GaLore [3] and FIRA [5] highlights the importance of performing optimization in the top eigenspace, and our method follows this established perspective rather than proposing an isolated design choice.
>
> For computational efficiency, we estimate the top eigenspace using power iteration, the same approach used by SOAP. In large-scale settings, power iteration is the only feasible option; full SVDs or alternative decompositions introduce prohibitive memory and compute costs, contradicting the goal of a lightweight optimizer. We have included ablations on the rank of the top eigenspace in Table 4.

---

> ### Author Response · Authors · 2025-11-21
>
> **Q1. Could you provide theoretical analysis showing when COSMOS converges faster than pure SOAP or MUON?**
>
> As noted earlier, neither SOAP nor MUON provides formal convergence guarantees or theoretical analysis showing improved convergence over AdamW. This is common among modern optimizer designs targeting LLMs, where the complexity of the models and training dynamics makes such theoretical understanding extremely challenging. We believe that developing such theory is an important open problem but would require substantial standalone effort. Our focus in this work is empirical: we test whether splitting optimization across eigenspaces yields practical benefits in large-scale training. We provide extensive experiments showing that COSMOS achieves better memory and speed efficiency while preserving optimization quality.
>
> ---
>
> **Q2. Have you explored adaptive rank selection based on the eigenvalue spectrum?**
>
> Adaptive rank selection is an interesting direction for future work. In the current paper, our goal is to demonstrate the advantage of the hybrid SOAP–MUON design. Experiments show that even with a fixed rank, COSMOS achieves strong and stable performance across models and datasets. Given the scale of our experiments, we believe the current study is already comprehensive and forms a good foundation for future investigations into adaptive rank strategies.
>
> ---
>
> **Q3. Could you test on at least one non-transformer architecture to demonstrate generalizability?**
>
> Please see our response to **Weakness 2**.
>
> ---
>
> **Q4. Is there a principled way to choose gamma without the eta/eta0 heuristic?**
>
> Our initial choice gamma = eta was empirical and worked well without tuning. In later experiments, we found that this mainly happens because the MUON learning rate eta is consistently about 0.1 to 0.5 times the base learning rate eta0 in the embedding layer, and gamma performs well across this range. We varied gamma in [0.1, 1] (see Table 4) and observed that COSMOS consistently outperforms MUON, showing robustness to gamma.
>
> In follow-up ablations, gamma = 0.25 often yields near-optimal performance. We re-ran Table 3 with gamma fixed at 0.25 and observed further improvement:
>
> **Validation perplexity under different learning rates (COSMOS, γ=0.25, r=64)**
>
> | lr     | 2e-4   | 5e-4   | 1e-3   | 2e-3   |
> |--------|-------|-------|-------|-------|
> | COSMOS | 21.09  | 20.54  | 20.57  | 20.87  |
>
>
> Therefore, gamma = eta/eta0 is not essential. In practice, setting gamma in [0.1, 0.5] is sufficient.
>
>
> ---
>
> ---
>
> **References**
>
> [1] Nikhil Vya et al. “SOAP: Improving and Stabilizing Shampoo using Adam”.
>
> [2] Keller Jordan et al. “Modded-nanogpt: Speedrunning the NanoGPT baseline”.
>
> [3] Jiawei Zhao et al. “GaLore: Memory-Efficient LLM Training by Gradient Low-Rank Projection”.
>
> [4] Huizhuo Yuan et al. “MARS: Unleashing the Power of Variance Reduction for Training Large Models”.
>
> [5] Xi Chen et al. “FIRA: Can We Achieve Full-rank Training of LLMs under Low-rank Constraint?”.
>
> [6] Kaplan et al. “Scaling Laws for Neural Language Models”.
>
> [7] Bello et al. “Revisiting ResNets: Improved Training and Scaling Strategies”.

---

### Author Response · Authors · 2025-12-03
**Summary for AC**

Dear AC,

We thank the reviewers for their thoughtful feedback and their engagement during the rebuttal period. For your convenience, we provide below a concise summary of the main points discussed:

---

**Reviewer NctS** noted in the official review that they wished to see experiments under additional settings. We first argued that the breadth of our current experiments already matches or exceeds prior work (SOAP, MUON, Galore, etc.). We then added the requested experiments, specifically setting the weight decay to 0.1 and switching to the higher-quality FineWeb dataset. **The reviewer acknowledged these additions and raised their score from 4 to 6.**

---

**Reviewer isQ9** raised several questions, including: (1) how $\gamma$ was chosen in COSMOS; (2) why replacing the top-eigenspace update in Muon with SOAP accelerates convergence; and (3) whether it would be interesting to study the effect of applying SOAP to different ranges of eigenspaces.

For (1), we explained that the choice was empirical and provided additional experiments showing that fixing $\gamma$ at 0.25 yields even better performance.

For (2), we emphasized that COSMOS combines the strengths of SOAP and MUON, leading to better per-iteration efficiency than either optimizer alone.

For (3), we agreed that this is an intriguing direction and noted that our work provides the foundation for such future exploration.
**Reviewer isQ9 accepted these points and raised their score from 4 to 6.**

---

**Reviewer vF7y** primarily expressed concerns about the profiling experiments for overhead. They questioned our gradient accumulation setting and requested a wall-clock time plot on NanoGPT. We argued that gradient accumulation is a commonly used technique in modern pretraining, and our setting was designed to simulate the real training time required for a 1B model on a single A100 node. We also provided the requested wall-clock plot on NanoGPT, showing that COSMOS still outperforms Muon. **Reviewer vF7y said that we had addressed these concerns and increased their score from 6 to 8.**

---

We regret that due to the OpenReview bug, two reviewers were unable to respond:

**Reviewer FdBR** originally gave a score of 6. They raised a question about rank selection; in response, we added experiments showing the singular value distribution of the gradients to illustrate the rationale behind our rank choice.

---

**Reviewer BuFv** initially gave a score of 4. Their questions included: (1) the choice of $\gamma$; (2) experiments on architectures beyond transformers (e.g., CNNs); (3) theoretical guarantees for COSMOS; and (4) adaptive rank selection in COSMOS.

For (1), our response matched that given to **Reviewer isQ9**: we added experiments fixing $\gamma$ at 0.25.

For (2), we explained that transformers are currently the only architecture with strong evidence of effective scaling; CNNs and LSTMs lack such scaling trends. For small-scale models, the training cost is already low, so a more efficient optimizer is less critical. Our intended scope is thus LLM optimization—clearly stated in both the title and abstract—and consistent with prior optimizer work (SOAP, Muon, Galore).

For (3), we noted that a complete theoretical treatment for Muon, SOAP, Shampoo, or even AdamW remains beyond the current understanding of the deep learning community, and also beyond the scope of an empirical paper such as ours. Consequently, providing a full theoretical analysis for COSMOS—being a hybrid optimizer built upon SOAP and MUON—would be equally challenging. Instead, we offer extensive empirical evidence demonstrating the superiority of COSMOS.

For (4), we agreed that adaptive rank selection is an interesting direction and leave it to future work. Our current work provides the foundation for pursuing this direction.

---

These discussions remain visible on the page, and the rating changes are fully verifiable since each reviewer noted their update when editing their review. Thank you for taking the time to read the official reviews and the associated discussions.

---

### Meta-Review · Area_Chair_SFwR · 2026-01-06

**Summary:**

Reviewers generally agreed that COSMOS presents a novel and practically meaningful hybrid optimizer that combines the strengths of SOAP and MUON via eigenspace decomposition, achieving substantial memory savings while maintaining competitive optimization performance for large language model pretraining. The idea of applying second-order adaptation only in a low-dimensional leading eigenspace was widely viewed as intuitive, well-motivated, and potentially impactful. The main concerns were about experimental scope, clarity of motivation, and practical validation, most of them were addressed during rebuttal.

**Reviewer Concerns:**

**Concerns Addressed**

Expanded experimental validation: added new experiments

Wall-clock and profiling clarity: concerns about gradient accumulation and step-time dominance were addressed, with detailed explanations of realistic training regimes and new wall-clock comparisons. Reviewer vF7y confirmed that these concerns were resolved and raised their score to 8.

Hyperparameter robustness and rank selection:
The authors provided joint grid searches over rank r and γ, additional ablations, and empirical evidence (including gradient singular value spectra) justifying fixed low-rank choices. This resolved rank-selection concerns raised by FdBR and BuFv.

Design motivation and intuition: The rebuttal clarified why combining SOAP in the well-estimated top eigenspace with MUON in the residual space improves stability and efficiency, addressing earlier doubts about the marginality of gains over MUON alone.

Concerns Still Outstanding

Formal theoretical guarantees: Some reviewers (BuFv and FdBR) would still prefer stronger theoretical justification or convergence analysis. However, these concerns were largely reframed as future-work directions.

Broader architectural generality: The method is evaluated only on transformer-based models. While reviewers accepted this as appropriate given the paper’s stated scope (LLM pretraining), broader generalization remains an open direction.

**Reviewer Scores:**

Reviewer NctS (Initial 4): raised to 6 after additional FineWeb, weight decay, overtraining, and batch-size experiments.

Reviewer isQ9 (Initial 4): raised to 6 after clarification of algorithm design, γ selection, and added ablations.

Reviewer vF7y (Initial 6): raised to 8 after wall-clock profiling and NanoGPT experiments addressed efficiency concerns.

Reviewer FdBR (Initial 6): Given that their rank-selection concern was directly addressed with new spectral analyses, it is reasonable to expect their score would remain at least 6.

Reviewer BuFv (Initial 4): Given that all four of their major concerns were addressed or contextualized, may increase to 6.

---

### Decision · Program_Chairs · 2026-01-26

Accept (Poster)